# The effect of human immunodeficiency virus infection on adverse events during treatment of drug-resistant tuberculosis: A systematic review and meta-analysis

Gilbert Lazarus[1]*, Kevin Tjoa[1], Anthony William Brian Iskandar[1], Melva Louisa[2], Evans L. Sagwa[3], Nesri Padayatchi[4], Vivian Soetikno[2]*

**1** Faculty of Medicine, Universitas Indonesia, Jakarta, Indonesia, **2** Department of Pharmacology and Therapeutics, Faculty of Medicine, Universitas Indonesia, Jakarta, Indonesia, **3** Independent Pharmacoepidemiologist, Windhoek, Namibia and Nairobi, Kenya, **4** CAPRISA MRC-HIV-TB Pathogenesis and Treatment Research Unit, Durban, South Africa

* gilbert.lazarus@ui.ac.id (GL); vivian.soetikno@ui.ac.id (VS)

**Data Availability Statement:** All relevant data are within the manuscript and its Supporting information files.

## Abstract

### Background

Adverse events (AEs) during drug-resistant tuberculosis (DR-TB) treatment, especially with human immunodeficiency virus (HIV) co-infection, remains a major threat to poor DR-TB treatment adherence and outcomes. This meta-analysis aims to investigate the effect of HIV infection on the development of AEs during DR-TB treatment.

### Methods

Eligible studies evaluating the association between HIV seropositivity and risks of AE occurrence in DR-TB patients were included in this systematic review. Interventional and observational studies were assessed for risk of bias using the Risk of Bias in Nonrandomized Studies of Intervention and Newcastle-Ottawa Scale tool, respectively. Random-effects meta-analysis was performed to estimate the pooled risk ratio (RR) along with their 95% confidence intervals (CIs).

### Results

A total of 37 studies involving 8657 patients were included in this systematic review. We discovered that HIV infection independently increased the risk of developing AEs in DR-TB patients by 12% (RR 1.12 [95% CI: 1.02–1.22]; $I^2 = 0\%$, p = 0.75). In particular, the risks were more accentuated in the development of hearing loss (RR 1.44 [95% CI: 1.18–1.75]; $I^2 = 60\%$), nephrotoxicity (RR 2.45 [95% CI: 1.20–4.98], $I^2 = 0\%$), and depression (RR 3.53 [95% CI: 1.38–9.03]; $I^2 = 0\%$). Although our findings indicated that the augmented risk was primarily driven by antiretroviral drug usage rather than HIV-related immunosuppression, further studies investigating their independent effects are required to confirm our findings.

**Funding:** This project was funded by the PUTI Q1 research grant from the Universitas Indonesia (grant number: BA-1005/UN2.RST/PPM.00.03.01/2020). The funder had no role in study design, data collection and analysis, decision to publish, or preparation of the manuscript.

**Competing interests:** The authors have declared that no competing interests exist.

## Conclusion

HIV co-infection independently increased the risk of developing AEs during DR-TB treatment. Increased pharmacovigilance through routine assessments of audiological, renal, and mental functions are strongly encouraged to enable prompt diagnosis and treatment in patients experiencing AEs during concomitant DR-TB and HIV treatment.

## Introduction

Drug-resistant tuberculosis (DR-TB), defined as the emergence of resistance pattern of TB bacilli to one or more anti-TB drugs [1], remains a major global health burden with approximately 465,000 cases of rifampicin-resistant/multidrug-resistant tuberculosis (RR/MDR-TB) and 1.4 million cases of isoniazid-resistance TB cases in 2019. Moreover, about 182,000 deaths from RR/MDR-TB were also reported in the same year [2]. This alarming evidence is further aggravated by the fact that DR-TB patients are more susceptible to drug-related adverse events (AEs) when compared to drug-susceptible TB patients [3], indicating that better understanding on the factors associated with the development of AEs during DR-TB treatment is urgently needed. This is saliently important, considering that AEs remain as one of the major predictors of unfavorable treatment outcomes [4, 5].

In light of this, studies have also shown that human immunodeficiency virus (HIV) infection is also prevalent among DR-TB patients particularly in low- and middle-income countries [1, 6]. Several factors including HIV-induced immunosuppression, viral-mediated toxicity, and the possibilities of additional drug-drug interactions suggest a potential interplay between these two major infectious diseases in affecting treatment outcomes [7, 8]. This necessitates the importance of comprehending the effect of HIV infection on the development of AEs during DR-TB treatment to help clinicians anticipate and promptly treat these patients, hence preventing further deterioration of treatment adherence and outcomes. As the current evidence remains equivocal [6, 9], this systematic review aims to thoroughly investigate the effect of HIV infection on AE occurrence during DR-TB treatment.

## Materials and methods

This review was conducted based on the guideline of systematic review of prognostic factor studies guideline proposed by Riley et al. [10] and was reported according to the Preferred Reporting Items for Systematic Reviews and Meta-Analyses (PRISMA) statement [11]. A detailed protocol has been prospectively registered in PROSPERO (CRD42020185029 [12]). Deviations from the protocol are described in **S1 Table in** S1 File.

### Search strategy

Literature searches were performed systematically by screening for eligible studies published up to 2 October 2020 through PubMed, Scopus, Cochrane Controlled Register of Trials (CENTRAL), MEDLINE (via EBSCO), and Cumulative Index to Nursing and Allied Health Literature (CINAHL) databases, using keywords listed on **S2 Table in** S1 File. Additional searches for grey literature were conducted through Google Scholar and ProQuest databases, in addition to literature snowballing of references from included studies and similar reviews. Screening and searching of relevant studies were conducted by two independent investigators (GL and KT), and any discrepancies were resolved by a third investigator (ML) in a blinded

fashion. No language restrictions were applied, and any title or abstracts deemed potentially eligible for inclusion by either investigator were retrieved for full-text assessments.

## Study eligibility criteria

Studies were included in this review if they met the following inclusion criteria: (1) design, interventional or observational studies including but not limited to cohort, case-control, and cross-sectional studies; (2) studies enrolling both HIV-infected and HIV-uninfected DR-TB patients receiving second line anti-TB drugs; and (3) studies reporting AEs based on HIV sero-positivity. AEs were defined as any untoward event occurring following the administration of second-line anti-TB and/or antiretroviral (ARV) agents. Conversely, criteria for exclusion were: (1) non-original research, including qualitative research, case studies, reports, or case series with <20 patients; (2) irretrievable full-text articles; or (3) articles not in English.

Considering that the association between HIV infection and AE occurrence is usually studies as part of a greater cohort, the authors made every effort to minimize rejection of studies judged potentially eligible for inclusion but did not dichotomize AEs based on HIV seropositivity, and the corresponding authors of the respective studies were contacted to obtain the additional data. When no response was provided or the authors were unable to retrieve the data, the studies were excluded from this systematic review (see S1 File **pg. 8**).

## Data extraction and quality assessment

The following data were extracted from each included study: (1) first author's last name and year of publication; (2) patient recruitment period; (3) study characteristics, viz. study design, location, and exposures; (4) patient characteristics, viz. sample size, mean age, frequency and proportion of male patients, proportion of HIV-infected patients, and TB resistance type (**see** S1 File **pg. 6, 7**); and (5) outcomes related to AEs. The primary outcome of this study was the frequency of patients experiencing at least one AE, while the secondary outcomes were: (1) frequency of patients experiencing at least one serious AE (SAE) and (2) frequency of specific AEs–as classified according to the Division of AIDS (DAIDS) Table for Grading the Severity of Adult and Pediatric Adverse Events, Corrected Version 2.1 [13]. When the reported outcomes were not listed in the DAIDS catalogue, AEs were categorized as per the Common Terminology Criteria for Adverse Events (CTCAE) Version 5.0 [14], or per authors' definitions when the reported outcomes were not available in both DAIDS and CTCAE directories. A serious AE was defined as an AE leading to treatment suspension, withdrawal or discontinuation, requiring prolonged hospitalization or immediate interventions to prevent permanent damage, or resulting in significant disabilities, congenital abnormalities, or death [15]. In the case of studies reporting the aforementioned outcomes only through graphical illustrations, the data were digitized with GetData Graph Digitizer ver. 2.26 (www.getdata-graph-digitizer. com).

The included studies were further assessed for methodological quality using the Risk of Bias in Non-randomized Studies of Intervention (ROBINS-I) for non-randomized studies of intervention (NRSI), Newcastle-Ottawa scale (NOS) [16] for cohort and case-control studies and the modified NOS tool for cross-sectional studies [17] (**S4 and S5 Tables in** S1 File, respectively). Studies assessed with NOS were subsequently classified as yielding low (0–3 stars), moderate (4–6 stars), or high (7–9 stars for longitudinal studies and 7–10 stars for cross-sectional studies) quality. Data extraction and bias assessments were conducted by two independent authors (KT and AW) and any discrepancies were resolved by a third author (GL)–also in an independent manner.

## Statistical analysis

Meta-analysis was performed with R ver. 4.0.0 (R Foundation for Statistical Computing, Vienna, Austria) [18] with the additional *meta* (ver. 4.9–6) [19], *metafor* (ver. 1.4–0) [20], and *robvis* [21] packages, while additional analyses were performed with MetaXL software ver. 5.3. (www.epigear.com) [22]. Both adjusted and unadjusted estimates were pooled in the meta-analysis; however, adjusted estimates were prioritized for the interpretation of the results [23]. As the covariates adjusted in each study are highly variable [10], we pre-specified a minimum adjustment factors of age and sex for study estimates to be included in the analysis.

We selected relative risk (RR) along with their 95% confidence interval (CI) as the common measure of the association between HIV infection and AEs. Odds ratios (ORs) and hazard ratios (HRs) were converted using the formula provided by the Cochrane Handbook ver. 6.0 [24] and VanderWeele et al. [25], respectively (**see** S1 File **pg. 10**). As clinical heterogeneity was anticipated, effect sizes were pooled using random-effects models [26]. Heterogeneity was investigated with Cochran Q statistics ($p < 0.10$ indicated statistical heterogeneity) and $I^2$ value–classified as negligible (0–25%), low (25–50%), moderate (50–75%), or high (>75%).

A priori, we determined subgroup and sensitivity analyses only for the pooled adjusted effects. Subgroup analyses were performed to address potential sources of heterogeneity by categorizing studies based on sample size, study design, location, and methodological quality; in addition to posteriori-determined subsets according to AE type, AE seriousness, and DR-TB treatment regimen. Furthermore, analysis based on study-reported effect size was also conducted to explore the impact of data conversion. We also performed additional analyses according to CD4 count, antiretroviral therapy (ART) status, TB resistance type, and AE severity to explore clinical disparities between subsets.

On the other hand, sensitivity analysis was performed by excluding studies with low methodological quality and sequentially omitting one study at a time. When the number of studies were adequate (n≥10) [27], potential publication bias were investigated visually by contour-enhanced funnel plot [28] and quantitatively by Egger's [29] and Begg's test [30]. In the case of detected publication bias, trim-and-fill analysis was performed to evaluate the potential sources of publication bias [31].

## Results

### Study selection and characteristics

Details of the literature search process are illustrated on Fig 1. The initial search retrieved 5316 records, of which 722 were deduplicated and 4496 were excluded following title and abstracts screening, resulting in the full-text assessments of 98 studies. Sixty-one studies were further excluded due to inappropriate settings (31 with insufficient information for reviewers to judge the availability of potential data, six did not include HIV-negative patients, and two did not report the number of HIV-negative patients experiencing AEs), no reported outcome of interests (14 studies), inadequate sample size (three studies), incomprehensible language (three studies), and irretrievable full-text articles (two studies). Therefore, a total of 37 studies were included in this systematic review. Four studies had sufficiently distinct statistical methods; however, we discovered that the four studies involved the same cohort [32–35]–thus we decided to regard them as one study for the remainder of this article. Four studies were later excluded from the meta-analysis due to distinct study designs (one interventional trial [36] and one cross-sectional study [5]) and incompatible effect measures (i.e. only p-values were reported by two studies [37, 38]). Among the included studies (i.e. 34 studies), 32 were cohort studies (22 retrospective and 10 prospective) and one each was a cross-sectional and an

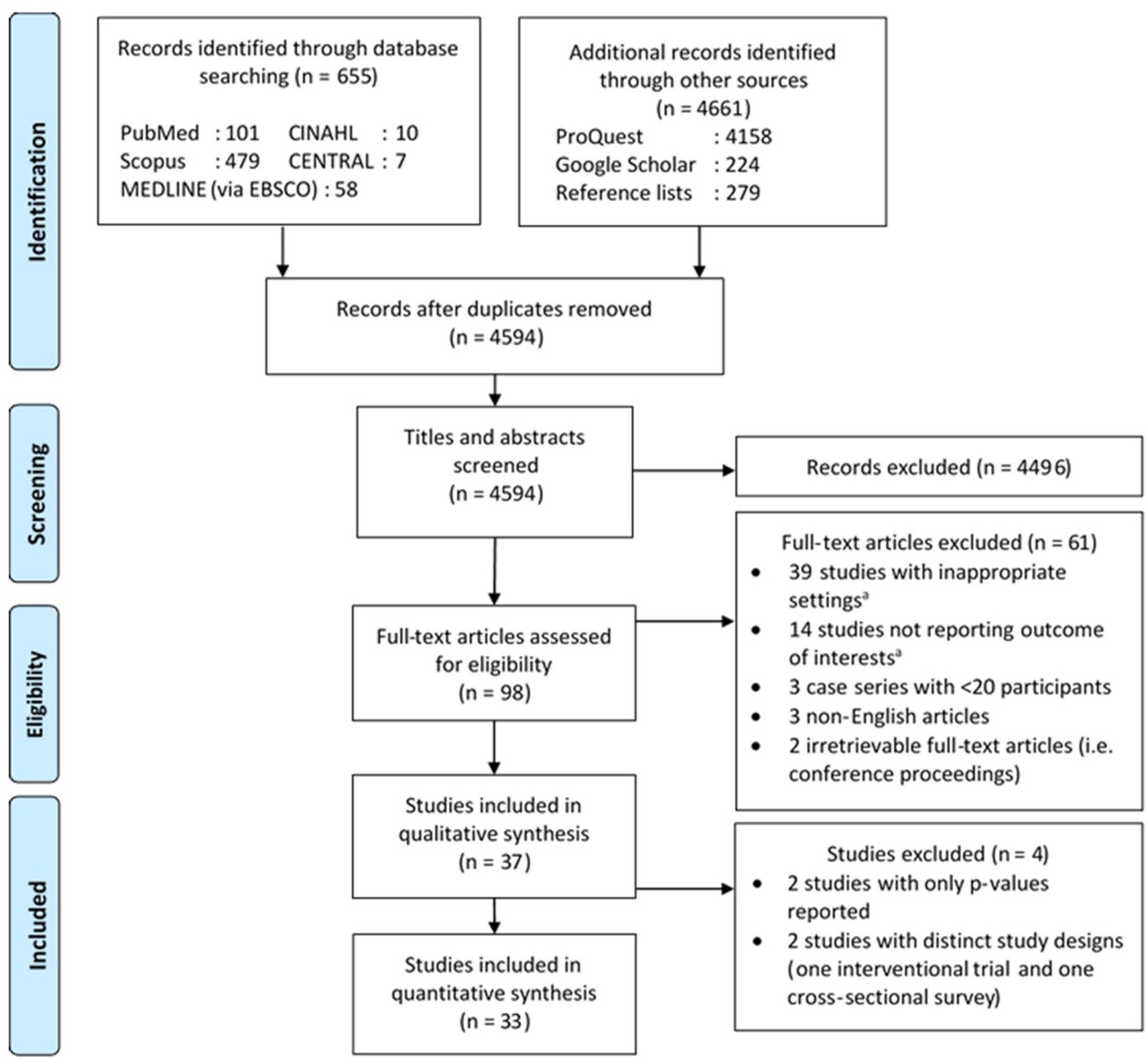

**Fig 1. Diagram flow illustrating the literature search process.** [a]See S1 File pg. 8 for further details. CENTRAL, Cochrane Controlled Register of Trials; CINAHL, Cumulative Index to Nursing and Allied Health Literature.

interventional study. Most of the included studies were conducted in Africa (30 studies), while the other two each in Asia and America (**S6 Table in** S1 File).

A total of 8657 patients (range: 27–1390) were included in this review, where 4437 (51.3%) were co-infected with HIV and 2832 (63.8%) were concomitantly treated with ARV drugs. With regards to TB resistance pattern, 7263 patients were diagnosed with MDR-TB, while 605 with XDR-TB. Pyrazinamide was the most extensively used anti-TB drugs (6190 patients, 25 studies), followed by ethionamide (4289 patients, 23 studies) and kanamycin (4241 patients, 23 studies). On the other hand, lamivudine (897 patients, 11 studies), efavirenz (862 patients, 15 studies), and tenofovir (830 patients, 13 studies) were the most widely used antiretroviral drugs. However, ART regimens were not consistently reported in the included studies, hence potentially underestimating the observed prevalence.

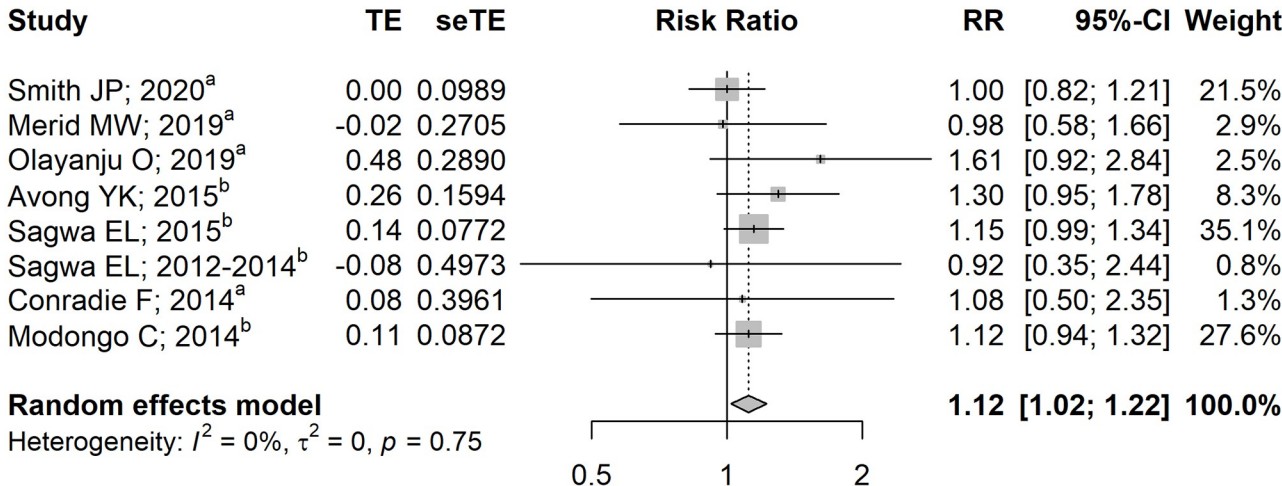

**Fig 2. Pooled adjusted effects on the association between HIV co-infection and the occurrence of adverse events.** [a]Effect estimate calculated using the formula provided by Cochrane Handbook ver. 6.0 [24]; [b]Effect estimate calculated using the formula provided by VanderWeele et al. [25] **CI**, confidence interval; **RR**, risk ratio.

Quality assessments of the observational studies revealed a predominant moderate-to-high methodological quality (8 and 24 studies, respectively; **S7, S8 Tables in** S1 File), while the only NRSI study yielded a serious risk of bias (**S1 Fig in** S1 File). All studies controlled for age or sex, and most studies satisfied the criteria for the representativeness of exposed and non-exposed cohorts (30 and 31 studies, respectively). On the other hand, half of the included studies failed to demonstrate that outcome of interest was not present at baseline, and 12 studies did not sufficiently adjust for additional confounders. These limitations may potentially be explained by the fact that most of the included studies were retrospective, thus predisposing these studies to selection bias [39].

## Outcomes

Potential overlapping populations were observed in ten studies [40–49] (**S9 Table in** S1 File), of which Hong et al. [43], Perumal et al. [40], Sagwa et al. [48], and Modongo et al. [49] were prioritized for analyses due to larger sample sizes. We discovered that HIV co-infection independently increased the risk of developing AE by 12% with negligible heterogeneity (RR 1.12 [95% CI: 1.02–1.22]; $I^2$ = 0%, p = 0.75; Fig 2), which was consistent with our findings on the pooled unadjusted effects (RR 1.18 [95% CI: 1.05–1.32]; $I^2$ = 62%, p<0.01; **S2 Fig in** S1 File). Subgroup analyses revealed no substantial heterogeneity between subgroups (Table 1), thus further ascertaining the consistency of our model. Furthermore, our model remained robust following sensitivity analysis (**S3 Fig in** S1 File), except when Sagwa et al. [48] or Avong et al. [50] was excluded, which resulted in borderline significance. In addition, we also discovered that injectable DR-TB regimens resulted in a more apparent risk of AE than all-oral DR-TB regimens (RR 1.11 [95% CI: 1.01–1.21] vs RR 1.61 [95% CI: 0.92–2.84]; Table 1), although further studies are needed to confirm our premises due to paucity of studies in the all-oral subgroup.

As we were unable to perform analyses to explore clinical disparities between subgroups on the adjusted model, we decided to perform these analyses on the unadjusted model. Unsurprisingly, we discovered that the risk of developing AE was more consistent in patients receiving ART. Moreover, we also discovered that the risk was more pronounced in MDR-TB

**Table 1. Summary of meta-analysis and subgroup analyses for the pooled adjusted effects on the association between HIV co-infection and adverse events occurrence.**

| Outcome | Studies | Events/N | | Outcome | | Heterogeneity | |
|---|---|---|---|---|---|---|---|
| | | HIV+ | HIV- | RR (95% CI) | P | I² | P |
| Adverse events | 8 | 772/1333 | 554/1049 | 1.12 (1.02–1.22) | 0.014 | 0% | 0.75 |
| *Subgroup analysis* | | | | | | | |
| AE type | | | | | 0.748 | | |
| Any AE | 5 | 427/680 | 367/676 | 1.10 (0.94–1.28) | | 2% | 0.39 |
| Specific AE | 3 | 345/6533 | 187/373 | 1.13 (1.01–1.27) | | 0% | 0.97 |
| AE seriousness | | | | | 0.618 | | |
| Any AE | 7 | 566/918 | 468/894 | 1.12 (1.03–1.23) | | 0% | 0.67 |
| Serious AE | 1 | 206/415 | 86/155 | 0.98 (0.58–1.67) | | NA | NA |
| ES type | | | | | 0.354 | | |
| HR | 4 | 433/802 | 167/283 | 1.05 (0.88–1.24) | | 0% | 0.47 |
| OR | 4 | 339/531 | 387/766 | 1.15 (1.03–1.28) | | 0% | 0.82 |
| Sample size | | | | | 0.360 | | |
| <100 patients | 2 | 61/68 | 46/52 | 1.40 (0.86–2.29) | | 0% | 0.33 |
| ≥100 patients | 6 | 711/1265 | 508/997 | 1.11 (1.01–1.22) | | 0% | 0.78 |
| Location | | | | | 0.488 | | |
| South Africa | 4 | 412/675 | 166/277 | 1.09 (0.96–1.23) | | 0% | 0.44 |
| Others | 4 | 360/658 | 388/772 | 1.16 (1.02–1.32) | | 0% | 0.77 |
| Design | | | | | 0.781 | | |
| Prospective | 3 | 227/387 | 81/128 | 1.10 (0.85–1.42) | | 19% | 0.29 |
| Retrospective | 5 | 545/946 | 473/921 | 1.14 (1.03–1.27) | | 0% | 0.87 |
| TB resistance type | | | | | 0.189 | | |
| MDR-TB | 5 | 428/716 | 348/690 | 1.10 (0.97–1.23) | | 0% | 0.72 |
| XDR-TB | 1 | 33/37 | 23/26 | 1.61 (0.92–2.84) | | NA | NA |
| DR-TB treatment regimen | | | | | 0.199 | | |
| Injectables | 7 | 739/1296 | 531/1023 | 1.11 (1.01–1.21) | | 0% | 0.85 |
| All-oral | 1 | 33/37 | 23/26 | 1.61 (0.92–2.84) | | NA | NA |

**AE**, adverse event; **CI**, confidence interval; **ES**, effect size; **HIV**, human immunodeficiency virus; **HR**, hazard ratio; **MDR-TB**, multidrug-resistant tuberculosis; **N**, total sample size; **NA**, not available; **OR**, odds ratio; **RR**, risk ratio; **TB**, tuberculosis; **XDR-TB**, extensively drug-resistant tuberculosis.

patients (RR 1.22 [95% CI: 1.04–1.42] vs XDR-TB: RR 0.80 [95% CI: 0.51–1.26]; **S4 Fig in** S1 File). Assessment of publication bias revealed a symmetrical funnel plot (**S5 Fig in** S1 File), which was further ascertained by non-significant Egger's and Begg's tests (p = 0.160 and p = 0.275, respectively). As no publication bias was detected, a trim-and-fill analysis was not performed. Publication bias assessment of the adjusted model was not conducted due to study paucity (n<10).

Contrary to our findings on the primary outcome, we discovered that HIV co-infection was not associated with the development of serious AE (RR 0.93 [95% CI: 0.58–1.50]; Fig 3). However, it is worth noting that these effects were unadjusted for confounders as we were unable to perform a formal analysis on the adjusted effects due to paucity of studies. Nonetheless, Merid et al. further ascertained our findings, stating that HIV co-infection was not an independent risk factor of serious AE occurrence [51]. Similarly, we discovered that the extent of immunosuppression was not associated with risks of developing AE (**S4 Fig in** S1 File), an observation

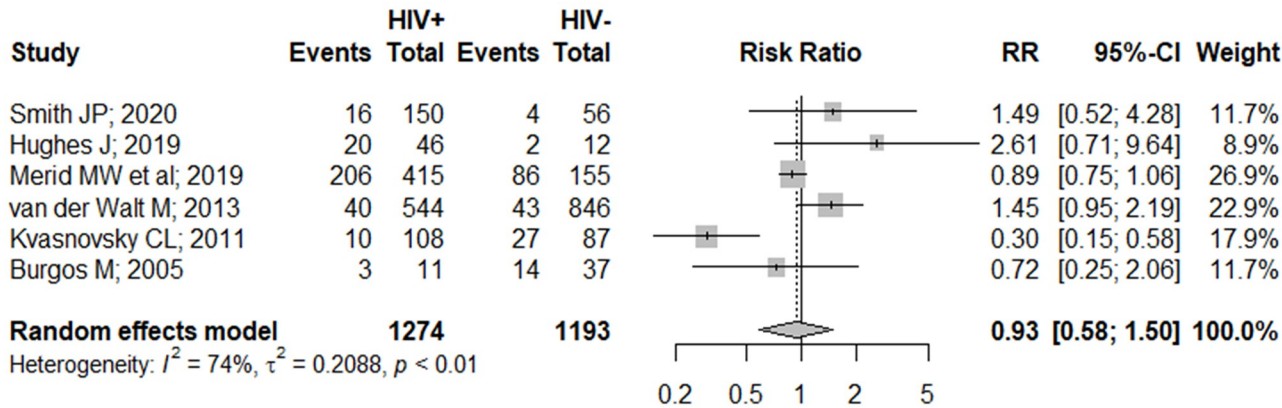

**Fig 3. Pooled unadjusted effects on the association between HIV co-infection and the occurrence of serious adverse events. CI**, confidence interval; **RR**, risk ratio.

which was further supported by Smith et al. who found that severe immunosuppression (i.e. CD4 count <200 cells/mm$^3$) was not independently associated with increased AE risk [44].

When analysis was conducted for specific AEs, we discovered that HIV co-infection increased the risk of hearing loss (RR 1.44 [95% CI: 1.18–1.75]), depression (RR 3.53 [95% CI: 1.38–9.03]), and renal impairment (RR 2.45 [95% CI: 1.20–4.98], **S10 Table in** S1 File). Our findings on depression and renal impairment yielded negligible heterogeneity (both I$^2$ = 0%), although the models were relatively imprecise. On the other hand, our findings on hearing loss yielded moderate heterogeneity (I$^2$ = 60%; p<0.01). None of the other specific AEs yielded significant results, except for peripheral neuropathy which resulted in borderline significance (RR 1.49 [95% CI: 0.98–2.28]).

## Discussion

This meta-analysis revealed that HIV co-infection was independently associated with AEs in DR-TB patients. Although our findings were in contrast with the findings Schnippel et al. [6], the study interpreted the association between HIV infection and drug-related AEs using vote counting method based on subjective rules, thus potentially predisposing such an analysis to poor performance validity [27].

Further analyses revealed that the risk of developing AE was more accentuated in HIV/MDR-TB co-infected patients. It is plausible that the effect of HIV infection in XDR-TB patients may potentially be masked by other factors, including bacterial load and type of TB drugs [4, 52]. Shean et al. stated that capreomycin was accountable for almost half of drug-related treatment discontinuations in XDR-TB patients [4], which was further supported by our findings where capreomycin was the most prescribed drugs in treating XDR-TB [4, 53, 54]. Nonetheless, Lan et al. stated that capreomycin was relatively safer than amikacin or kanamycin for MDR-TB treatment [55], thus indicating that future studies directly comparing the use of capreomycin between MDR-TB and XDR-TB patients are required to confirm our findings. In addition, we also observed that HIV/DR-TB patients receiving concomitant ART were more vulnerable to AEs. Interestingly, we found that the extent of immunosuppression was not associated with AE occurrence, implying that the observed risk was presumably driven by drug-drug interactions rather than HIV-mediated immunosuppression. However, we were unable to explore the suspected drug-drug interactions as patient-level data were not available. Some anti-DR-TB drugs have been reported to induce adverse interactions when

concomitantly administered with ART, including bedaquiline, delamanid, and moxifloxacin, which had documented interactions with several protease inhibitors (e.g. lopinavir/ritonavir) and non-nucleoside reverse transcriptase inhibitors (i.e. efavirenz and rilpivirine) [56], thus potentially leading to multiple toxicities resulting in deleterious health consequences [57].

This meta-analysis further indicated that the effect of HIV infection was more prominent in the increased risk of developing hearing loss and nephrotoxicity. It has been well-established that some anti-TB drugs, particularly aminoglycosides, are associated with hearing loss [58, 59]. Aminoglycosides were also associated with nephrotoxicity as they were primarily renally excreted and may cause tubular necrosis [60, 61]. This is in line with our findings where amikacin and kanamycin were among the most utilized drugs in the included studies, thus further elaborating the observed link. In addition, Lan et al. also stated that these drugs resulted in the highest incidence of adverse events leading to permanent drug discontinuation [55], indicating that a safer and better-tolerated regimen is required to reduce aminoglycoside-related morbidities. In this regard, all-oral treatment regimens become the preferred option for most DR-TB patients [62]. This is saliently true considering that our findings suggest that injectable-containing DR-TB regimens may result in a more apparent risk of developing AE, although further studies are required to confirm these premises. Furthermore, all-oral DR-TB regimens have shown to be more cost-effective with less logistical challenges [63, 64], thus rendering them more worthwhile to be implemented especially in resource-limited settings. All in all, these necessitates the widespread implementation of oral-only regimens as the mainstay of DR-TB treatment.

Coincidentally, previous studies have postulated that antiretroviral drugs may also induce nephrotoxicity [40, 47], hence implying that both antiretroviral and antituberculosis drugs may synergistically exacerbate the risk of developing drug-related nephropathy [47]. Moreover, HIV infection has been presumed to yield a direct effect on hearing loss where it may cause severe immune dysfunction and inflammation resulting in loss of hearing functions [65]. This association has been demonstrated by previous studies where hearing loss in HIV patients were primarily driven by disease progression, independent of history of ARV medication [65, 66]. In addition to nephrotoxic and ototoxic effects, we also discovered that HIV co-infection may predispose DR-TB patients to depression. Although Das et al. stated that these symptoms may be pertinently improved following medications, it is imperative for these conditions to be promptly identified and treated as psychiatric illnesses have been associated with poor DR-TB outcomes. Furthermore, stringent drug selection and vigilance should be implemented as some anti-depressants have been documented to exhibit pharmacological interactions with ARV and anti-DR-TB drugs [67].

Altogether, although we discovered that HIV co-infection was associated with an increased risk of developing any AE in DR-TB patients, ART should not be needlessly deferred in such patients. This is especially true considering that HIV co-infection was not associated with an increased risk of serious AE occurrence. Rather, we encourage clinicians to increase pharmacovigilance on HIV/DR-TB co-infected patients, especially in terms of ototoxicity, nephrotoxicity, and depressive symptoms. Therefore, routine audiological, laboratory (i.e. renal panel), and mental health assessments on such patients are strongly recommended. These routine assessments should be performed periodically by taking into account the common onset of each AEs. However, our current data did not permit such an analysis due to differences in follow-up duration, thus limiting our ability to explore these factors. According to a study by Zhang et al. [68], most AEs in patients receiving injectable-containing DR-TB regimens occurred within the first six months. In contrast, most AEs in all-oral regimens appeared to develop more quickly, ranging between two weeks to three months [69, 70]. Considering this, it is plausible for such assessments to be performed monthly, thus allowing the early detection and prompt

management of potential AEs [71]. This is particularly important as AEs were among the most common reasons leading to treatment non-adherence and failure in DR-TB patients [72].

This study has several limitations. Although our findings highlighted the unfavorable effect of HIV co-infection on AE development in patients receiving DR-TB treatments, some of the pooled estimates were unadjusted for confounders due to study scarcity, emphasizing caution in the interpretation of our findings. In addition, there is a need for further studies evaluating the independent effect of ART and CD4 count on AE occurrence in HIV/DR-TB patients, and studies comparing the effect of HIV infection across different spectrum of AE seriousness and severity. We were also unable to perform subgroup analysis by duration of follow-up due to heterogeneity in reporting, suggesting that a standardized reporting of follow-up duration in future studies are urgently needed. Furthermore, our study was also limited by the fact that some potentially important factors such as specific ARV/anti-DR-TB regimens and drug-drug interactions remained unexplored as patient-level data were not available. Lastly, although language bias may arise from our eligibility criteria, our study included a relatively large number of patients and only three non-English articles were excluded, suggesting that any language bias may be negligible. Despite this, it is worth noting that most of the studies were conducted in Africa, thus warranting further studies from other regions to ascertain the generalizability our findings across the globe.

## Conclusion

In conclusion, this meta-analysis adds to the growing body of evidence supporting the independent association between HIV co-infection and AEs in DR-TB patients. Furthermore, we also discovered that HIV co-infection was associated with a remarkably increased risk of developing hearing loss, nephrotoxicity, and depression, indicating that meticulous and routine assessments of these patients, especially in terms of audiological, renal, and mental health functions, are required to avoid compromising treatment adherence and outcomes. Although our findings suggested that the augmented risk of AE occurrence in HIV/DR-TB co-infected patients may have been primarily driven by ARV usage rather than HIV-related immunosuppression, further studies are required to confirm these premises, which were unadjusted for potential confounders.

## Supporting information

**S1 File. Supplementary materials.**
(DOCX)

**S2 File. PRISMA checklist.**
(DOC)

## Author Contributions

**Conceptualization:** Gilbert Lazarus, Anthony William Brian Iskandar, Vivian Soetikno.

**Data curation:** Gilbert Lazarus, Kevin Tjoa.

**Formal analysis:** Gilbert Lazarus.

**Funding acquisition:** Vivian Soetikno.

**Investigation:** Gilbert Lazarus, Kevin Tjoa, Anthony William Brian Iskandar.

**Methodology:** Gilbert Lazarus.

**Project administration:** Gilbert Lazarus, Vivian Soetikno.

**Resources:** Gilbert Lazarus, Melva Louisa, Evans L. Sagwa, Nesri Padayatchi.

**Software:** Gilbert Lazarus.

**Supervision:** Melva Louisa, Vivian Soetikno.

**Validation:** Melva Louisa, Vivian Soetikno.

**Visualization:** Gilbert Lazarus.

**Writing – original draft:** Gilbert Lazarus, Kevin Tjoa.

**Writing – review & editing:** Gilbert Lazarus, Melva Louisa, Evans L. Sagwa, Vivian Soetikno.

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
