## [Decision Letter · Decision Letter 0]

30 Dec 2020

PONE-D-20-36024

The effect of human immunodeficiency virus infection on adverse events during treatment of drug-resistant tuberculosis: a systematic review and meta-analysis

PLOS ONE

Dear Dr. Lazarus,

Thank you for submitting your manuscript to PLOS ONE. After careful consideration, we feel that it has merit but does not fully meet PLOS ONE’s publication criteria as it currently stands. Therefore, we invite you to submit a revised version of the manuscript that addresses the points raised during the review process.

Although it is an interesting manuscript, it lacks sufficient clarity and detail for it to be considered for publication in its current form. Specifically, these are the major criticisms from my evaluation:

The Introduction section did not provide sufficient detail about the rationale for the review in the context of what is already known.An explicit statement of the question addressed concerning participants, exposures, comparisons, outcomes, and study design (PECOS) is required in the main document.The authors did not indicate if a review protocol exists and where it can be accessed.There is a contradiction in the Methods section: firstly, they reported that "No language restrictions were applied" and then they stated that a criterion for exclusion was "articles not in English".Authors should be aware that for cross-sectional studies, a commonly reported measure of effect is the Odds Ratio (OR), but it is common for ORs to be wrongly interpreted as Risk Ratios. In the present manuscript, the authors use relative risk for cross-sectional studies. Thus, corrections are required.

Two reviewers have constructively detailed some other relevant issues. Please try to address all issues carefully in a revised manuscript.

We look forward to receiving your revised manuscript.

Kind regards,

Endi Lanza Galvão

Academic Editor

PLOS ONE

Journal Requirements:

Reviewers' comments:

Reviewer's Responses to Questions

**Comments to the Author**

1. Is the manuscript technically sound, and do the data support the conclusions?

Reviewer #1: Partly

Reviewer #2: Partly

2. Has the statistical analysis been performed appropriately and rigorously? 

Reviewer #1: Yes

Reviewer #2: N/A

3. Have the authors made all data underlying the findings in their manuscript fully available?

Reviewer #1: Yes

Reviewer #2: Yes

4. Is the manuscript presented in an intelligible fashion and written in standard English?

Reviewer #1: Yes

Reviewer #2: Yes

5. Review Comments to the Author

Reviewer #1: General comments

1. The chosen theme of study is topical and has important significance for policy makers as well as clinicians.

2. The study uses good statistical methods for data analysis, and results are structured well.

3. The results and conclusions are mostly in line with other published data.

Major comments - Results and conclusions

1. While HIV is understood to be a risk factor for aggravating some of the adverse events seen during the MDR-TB treatment, it would have been good if the role of specific regimen and drug-drug interactions in such patients was discussed. This becomes specifically relevant in light of evolving recommendations for newer regimen for MDR-TB.

2. Overall 3 AEs are highlighted throughout the narrative – hearing loss, depression and renal impairment. However, looking at table S9, there are several other adverse events for which RR was greater than those for these adverse events. There is also no mention of whether statistical significance of these associations was checked. Hence, the reason for prioritizing the mention of just 3 adverse events needs to be clarified.

3. It may also be worth mentioning here that in Table 1 in the main narrative, where meta-analysis and sub group analyses for the pooled adjusted effects on the association between HIV co-infection and adverse events occurrence, is presented the RR for any AE is more than 1 for HIV+ individuals while RR for serious AE is equivocal for HIV+ and HIV- individuals. However, the three commonly associated AEs, as above, are generally considered serious AE (specifically hearing loss and renal impairment).

4. It is also expected that ‘serious AE’ would be a subset of ‘any AE’. However, this doesn’t appear to be the case in analysis shown in Table 1. This needs to be clarified.

Other minor comments

1. Terms ‘adverse reaction’ and ‘adverse events’ seem to have been used interchangeably at some places. The former term may be used if causality assessment has been conducted.

2. Line 54 – ‘440,000 new cases and 150,000 deaths each year’. These figures are for RR/MDR-TB while the sentence starts with only DR-TB. DR-TB is an encompassing term for other forms/types of resistance as well.

3. Line 93-94 – ‘AEs were defined as any untoward event occurring following the administration of second-line anti-TB agents’. Please clarify is this also included AEs associated with ARV administration.

4. Table 1: ‘HR’ and ‘OR’ abbreviations in the ‘Outcome’ column need to be elaborated.

5. As per Table 1, the RR for adverse events seems more pronounced for XDR-TB patients but this doesn’t find mention in subsequent narrative (line 234).

6. In addition to monitoring for AEs, use of injection free regimen could be part of the recommendations because some of the common and serious AEs are associated with the injections being used for treatment of RR/MDR-TB

Reviewer #2: The authors have conducted a study that could provide valuable information regarding treatment of drug-resistant TB patients with HIV infection. As DR-TB treatment is highly variable and factors at both the individual patient and treatment centre level contribute to treatment outcomes, questions regarding this are hard to address with aggregate level meta-analysis. However, there is a lot of data collected in this study and I think some valuable evidence can be generated with a few more sensitivity analyses, of which I outline below.

Major:

Quality assessment:

1. Newcastle-Ottawa scale is not a good tool and for these study designs involving an intervention (or in general: see Stang et al. 2018 Eur J Epidemiol. “Case study in major quotation errors: a critical commentary on the Newcastle–Ottawa scale”). I think the quality assessment should be redone using ROBINS-I tool or similar to assess bias. Additionally, a sensitivity analysis stratifying on study quality as done with the NOS grading should be included as well – as selection bias, duration of treatment, and timing of AE may all affect the point estimates presented in the included studies. Issues of duration of TB drug use and time to adverse event are important for interpretation of effect estimates.

Analysis:

2. On the conversion of OR/HR to RR. I am not convinced this method is appropriate for pooling effect estimates in a meta-analysis. This conversion method may still introduce bias. A sensitivity analysis where original odds ratios are pooled separately from risk ratios may be helpful to see if the pooled effects are similar across the effect measures used. In addition, it would be useful to indicate on any forest plots, in which they appear, the studies that have had their OR/HR converted.

Further, was there an a priori adjustment set for which you would considered appropriate to pool adjusted effect estimates? A mix of estimated effects that may have been adjusted for different covariates may not be appropriate either (especially if they are being converted). Some clear mention of this should be included in the methods as well.

3. One of the main predictors for AE are the specific drugs used (as mentioned in the discussion). Some stratification for cohorts that used PZA, Eto, and Km vs. those who did not (if possible given the data) should be attempted. As individual adjustments for drug use are not possible, this may help determine whether HIV is independently associated with AE rather than driven by the drugs used in specific cohorts.

4. In addition to the use of individual drugs, the use and availability of drugs has changed over the last two decades, as well as the guideline recommendations for DR-TB treatment. Combining cohorts from these different eras may not be appropriate as some assumptions between their comparability may be violated. Was the effect of HIV consistent between cohorts ending before 2010 and cohorts that began after 2010? Some stratification of cohort periods may provide some insight into this.

If the data does not allow for these stratification to be made, some discussion regarding treatment periods in addition to the drugs used should be included.

Discussion:

5. Line 267 – this statement regarding what was reported in Lan et al. is not correct. In fact, what was reported for drug stoppage, if anything, was close to indicating the opposite (HIV may decrease the odds of having a drug stopped). This statement should be revised to reflect this.

6. As mentioned above: Issues of duration of TB drug use and time to adverse event are important for interpretation of effect estimates, and there should be some mention of this in the discussion.

Minor:

Line 57 – ‘Adverse event’ is first mentioned, but not abbreviated until line 64

Materials/methods – was a protocol registered/published?

Line 85 – “No language restrictions were applied”, but on line 96 an exclusion criteria is ‘articles not in English’, can you please clarify language restrictions?

Line 120 – I am not sure what is meant, do you mean data was extracted/estimated from figures if unavailable from text/tables? Some clarification would be helpful.

6. PLOS authors have the option to publish the peer review history of their article (what does this mean?). If published, this will include your full peer review and any attached files.

Reviewer #1: **Yes: **Vineet Bhatia

Reviewer #2: No

---

## [Author Response · Author response to Decision Letter 0]

29 Jan 2021

January 30th, 2021

Dear Prof. Joerg Heber, Dr. Endi Lanza Galvão, and respected reviewers,

Thank you for the comments from the editor and reviewers on our manuscript entitled “The effect of human immunodeficiency virus infection on adverse events during treatment of drug-resistant tuberculosis: a systematic review and meta-analysis” by Gilbert Lazarus, Kevin Tjoa, Anthony William Brian Iskandar, Melva Louisa, Evans L. Sagwa, Nesri Padayatchi, Vivian Soetikno (manuscript ID: PONE-D-20-36024). We really appreciate the constructive and detailed feedback on our manuscript. We have revised the current submitted manuscript based on the reviewers’ feedback. Please find the details of revisions attached within the 'Response to Reviewers' file in the submission system. 

Responses to In-house Editor

1. The Introduction section did not provide sufficient detail about the rationale for the review in the context of what is already known.

Thank you very much for the feedback. Regarding the critique to provide more details on the rationale for conducting the systematic review, we have added an additional explanation illustrating our reasoning on the potential interplay between HIV infection and drug-resistant tuberculosis (DR-TB) in the development of adverse events (AEs).

"In light of this, studies have also shown that human immunodeficiency virus (HIV) infection is also prevalent among DR-TB patients particularly in low- and middle-income countries[1,2]. Several factors including HIV-induced immunosuppression, viral-mediated toxicity, and the possibilities of additional drug-drug interactions suggest a potential interplay between these two major infectious diseases in affecting treatment outcomes.[3,4] This necessitates the importance of comprehending the effect of HIV infection on the development of AEs during DR-TB treatment to help clinicians anticipate and promptly treat these patients, hence preventing further deterioration of treatment adherence and outcomes. As the current evidence remains equivocal[2,5], this systematic review aims to thoroughly investigate the effect of HIV infection on AE occurrence during DR-TB treatment." [see page 4 line 64-73]

In short, we presumed that HIV infection, which may induce immunosuppression and toxicity to the host, may increase the likelihood of developing AEs in the treatment of DR-TB. Furthermore, as HIV-infected patients receive antiretroviral (ARV) treatments, these agents may induce drug-drug interactions either with the other ARV drugs or with anti-DR-TB drugs. Lastly, as the current evidence on the effect of HIV infection on the development of AEs in DR-TB patients remain equivocal[2,5], we decided to perform a systematic review to investigate this matter.

References:

1. Singh A, Prasad R, Balasubramanian V, Gupta N. Drug-resistant tuberculosis and HIV infection: current perspectives. HIV/AIDS (Auckland). 2020;12: 9–31. doi:10.2147/HIV.S193059

2. Schnippel K, Firnhaber C, Berhanu R, Page-Shipp L, Sinanovic E. Adverse drug reactions during drug-resistant TB treatment in high HIV prevalence settings: A systematic review and meta-analysis. J Antimicrob Chemother. 2017;72: 1871–1879. doi:10.1093/jac/dkx107

3. Naif HM. Pathogenesis of HIV infection. Infect Dis Rep. 2013;5: 26–30. doi:10.4081/idr.2013.s1.e6

4. Montessori V, Press N, Harris M, Akagi L, Montaner JSG. Adverse effects of antiretroviral therapy for HIV infection. CMAJ. Canadian Medical Association; 2004. pp. 229–238. 

5. Companion handbook to the WHO guidelines for the programmatic management of drug-resistant tuberculosis. Geneva: World Health Organization; 2014.

2. An explicit statement of the question addressed concerning participants, exposures, comparisons, outcomes, and study design (PECOS) is required in the main document. 

Thank you very much for the feedback. Regarding the critique to provide the PECOS of our study, we have added the PECOS in our eligibility criteria as below: 

"Studies were included in this review if they met the following inclusion criteria: (1) design, interventional or observational studies including but not limited to cohort, case-control, and cross-sectional studies; (2) studies enrolling both HIV-infected and HIV-uninfected DR-TB patients receiving second line anti-TB drugs; and (3) studies reporting AEs based on HIV seropositivity." [see page 5 line 96-100]

In this regard, the PECOS of our study is: (1) participants, DR-TB patients; (2) exposure, HIV infection; (3) comparison, no HIV infection; (4) outcome: AE occurrence; and (4) study design, interventional or observational studies. Furthermore, we have also added the detailed PICOTS (population, index prognostic factor, comparison prognostic factor, outcome, timing, and setting) of our study in Supplementary Table S2. We believe that the PICOTS framework is appropriate for prognostic reviews as the timing and setting of the prognostic factor studied (i.e. HIV infection) or the outcomes may define the focus of our review.[1] 

Reference:

Riley RD, Moons KGM, Snell KIE, Ensor J, Hooft L, Altman DG, et al. A guide to systematic review and meta-analysis of prognostic factor studies. BMJ. 2019;364. doi:10.1136/bmj.k4597

3. The authors did not indicate if a review protocol exists and where it can be accessed.

Thank you very much for the feedback. We have added our protocol to the manuscript, which was prospectively registered in PROSPERO on 5 July 2020.

"This review was conducted based on the guideline of systematic review of prognostic factor studies guideline proposed by Riley et al.[1] and was reported according to the Preferred Reporting Items for Systematic Reviews and Meta-Analyses (PRISMA) statement[2]. A detailed protocol has been prospectively registered in PROSPERO (CRD42020185029[3]). Deviations from the protocol are described in Supplementary Table S1." [see page 4-5 line 76-81]

In addition to citing our protocol, we have also specified the protocol in the PRISMA checklist (see Supplementary Material_PRISMA checklist). Furthermore, we have also added a supplementary table, listing all deviations made along with the explanations (see Supplementary Table S1). Following a thorough review, we discovered that interventional studies were not a criterion for exclusion, thus we decided to include the interventional study in our systematic review. The details of the revision are illustrated in Figure 1. As there is only one interventional study, we decided to exclude the study from the quantitative analysis (i.e., meta-analysis) and opted to analyze the study qualitatively. Details of the study’s characteristics and outcomes can be seen on Supplementary Table S6 and S9, respectively, while the result of risk of bias assessment of the aforementioned study are illustrated in Supplementary Figure S1. 

4. There is a contradiction in the Methods section: firstly, they reported that "No language restrictions were applied" and then they stated that a criterion for exclusion was "articles not in English".

Thank you very much for the feedback. Regarding the critique that there is a contradiction in the methods section concerning the eligibility criteria, we intended to assess whether language restrictions may affect our findings. Hence, we did not filter the search results by language, but rather excluded the potentially eligible non-English articles discovered during the search.

"Conversely, criteria for exclusion were: (1) non-original research, including qualitative research, case studies, reports, or case series with <20 patients; (2) irretrievable full-text articles; or (3) articles not in English." [see page 5-6 line 101-103]

The implementation of this criterion enabled us to assess the language bias arising from the language limitation, which we deemed negligible considering the large number of patients pooled in this systematic review and the fact that only three potentially eligible non-English articles were excluded. Regardless of this, we have acknowledged this as a limitation in the text as follows.

"Lastly, although language bias may arise from our eligibility criteria, our study included a relatively large number of patients and only three non-English articles were excluded, suggesting that any language bias may be negligible." [see page 19 line 356-359]

References:

1. Riley RD, Moons KGM, Snell KIE, Ensor J, Hooft L, Altman DG, et al. A guide to systematic review and meta-analysis of prognostic factor studies. BMJ. 2019;364. doi:10.1136/bmj.k4597

2. Moher D, Liberati A, Tetzlaff J, Altman DG, The PRISMA Group. Preferred reporting items for systematic reviews and meta-analyses: The PRISMA statement. PLoS Med. 2009;6: e1000097. doi:10.1371/journal.pmed.1000097

3. Lazarus G, Soetikno V, Iskandar A, Louisa M. The burden of human immunodeficiency virus infections on adverse events occurrence in the treatment of drug-resistant tuberculosis: a systematic review and meta-analysis. PROSPERO 2020. CRD42020185029. [cited 2021 Jan 19]. Available: https://www.crd.york.ac.uk/PROSPERO/display_record.php?RecordID=185029

5. Authors should be aware that for cross-sectional studies, a commonly reported measure of effect is the Odds Ratio (OR), but it is common for ORs to be wrongly interpreted as Risk Ratios. In the present manuscript, the authors use relative risk for cross-sectional studies. Thus, corrections are required.

Thank you very much for the feedback. We agree with the reviewer that the estimates of cross-sectional studies should not be interpreted as relative risks (RRs), but rather as odds ratios (ORs). In this study, there was only one cross-sectional study (i.e. Sineke et al[1]). Although the study expressed their outcomes with relative risks (RRs), we deemed that it is more appropriate to calculate the OR rather than RR, especially considering that the study is a cross-sectional survey. Furthermore, we obtained the outcome from binary data, thus enabling us to calculate the crude OR. 

However, we discovered that we inadvertently included the study in the analysis of unadjusted outcomes (i.e. primary outcome in Supplementary Figure S2 and subgroup analyses in Supplementary Figure S4). Hence, we decided to re-run the analysis with the exclusion of Sineke et al[1]. In short, no significant differences were found with the exclusion of Sineke et al[1], including the primary analysis (RR 1.18 [95% CI: 1.05-1.32]; Supplementary Figure S2) and antiretroviral therapy subgroup (RR 1.21 [95% CI: 1.02-1.43]; Supplementary Figure S4). However, as there were initially only two studies in the severe immunosuppression subgroup (CD4<50 and ≥50 cells/mm3), the exclusion of Sineke et al[1] refrained us from performing a meta-analysis. Hence, we decided to include the estimates in Supplementary Table S9 and qualitatively analyzed the evidence.

Reference:

1. Sineke T, Evans D, Schnippel K, van Aswegen H, Berhanu R, Musakwa N, et al. The impact of adverse events on health-related quality of life among patients receiving treatment for drug-resistant tuberculosis in Johannesburg, South Africa. Health Qual Life Outcomes 2019;17:94. https://doi.org/10.1186/s12955-019-1155-4.

Responses to Reviewer 1

1. While HIV is understood to be a risk factor for aggravating some of the adverse events seen during the MDR-TB treatment, it would have been good if the role of specific regimen and drug-drug interactions in such patients was discussed. This becomes specifically relevant in light of evolving recommendations for newer regimen for MDR-TB.

Thank you very much for the feedback. We agree with the reviewer that the role of specific regimen and drug-drug interaction may provide essential information in determining which DR-TB treatment regimen to use. However, as we did not intend to analyze patient-level data, and as the treatment regimens used in each study intertwined with each other, we were unable to analyze specific drug-drug interaction. In light of this, we discussed for a way to shed light on possible roles of specific regimen and drug-drug interaction and managed to perform a subgroup analysis based on DR-TB treatment regimen, dichotomizing studies into studies utilizing injectable agents (i.e. kanamycin, amikacin, and capreomycin) and all-oral regimens (i.e. using bedaquiline, pretomanid, and linezolid). We discovered that injectable-containing DR-TB regimens was associated with a more apparent risk of developing adverse event (RR 1.11 [95% CI: 1.01-1.21] vs all-oral regimens: RR 1.61 [95% CI: 0.92-2.84]; Table 1), although it is worth noting that further research are required to confirm our findings as there is currently limited evidence on studies utilizing all-oral DR-TB regimens. We have also added a discussion recommending the transition of injectable-containing DR-TB regimens to all-oral treatment regimens.

"In this regard, all-oral treatment regimens become the preferred option for most DR-TB patients.[1] This is saliently true considering that our findings suggest that injectable-containing DR-TB regimens may result in a more apparent risk of developing AE, although further studies are required to confirm these premises. Furthermore, all-oral DR-TB regimens have shown to be more cost-effective with less logistical challenges[2,3], thus rendering them more worthwhile to be implemented especially in resource-limited settings. All in all, these necessitates the widespread implementation of oral-only regimens as the mainstay of DR-TB treatment." [see page 17 line 309-316]

In addition, we have also shed light on the potential interplay between anti-DR-TB agents and ARV drugs in the development of treatment-related AE as proven by the increased risk of developing AE in HIV/DR-TB patients receiving ART (RR 1.21 [95% CI: 1.02-1.43] vs not on ART: RR 1.24 [95% CI: 0.90-1.69]; Table 1). Although we were unable to investigate specific drug-drug interaction between these drugs, we have discussed some potential drugs which have documented interactions.

"In addition, we also observed that HIV/DR-TB patients receiving concomitant ART were more vulnerable to AEs. Interestingly, we found that the extent of immunosuppression was not associated with AE occurrence, implying that the observed risk was presumably driven by drug-drug interactions rather than HIV-mediated immunosuppression. However, we were unable to explore the suspected drug-drug interactions as patient-level data were not available. Some anti-DR-TB drugs have been reported to induce adverse interactions when concomitantly administered with ART, including bedaquiline, delamanid, and moxifloxacin, which had documented interactions with several protease inhibitors (e.g. lopinavir/ritonavir) and non-nucleoside reverse transcriptase inhibitors (i.e. efavirenz and rilpivirine)[4], thus potentially leading to multiple toxicities resulting in deleterious health consequences[5]." [see page 16 line 290-299]

References:

1. World Health Organization. WHO consolidated guidelines on drug-resistant tuberculosis treatment 2019. Geneva: World Health Organization; 2019

2. Ionescu AM, Mpobela Agnarson A, Kambili C, Metz L, Kfoury J, Wang S, et al. Bedaquiline- versus injectable-containing drug-resistant tuberculosis regimens: a cost-effectiveness analysis. Expert Rev Pharmacoeconomics Outcomes Res 2018;18:677–89. https://doi.org/10.1080/14737167.2018.1507821.

3. Reuter A, Tisile P, Von Delft D, Cox H, Cox V, Ditiu L, et al. The devil we know: is the use of injectable agents for the treatment of MDR-TB justified? Int J Tuberc Lung Dis. 2017;21:1114–26

4. HIV Drug Interactions. Anti-tuberculosis treatment selectors. Univ Liverpool 2019. hiv-druginteractions.org/prescribing-resources (accessed October 11, 2020).

5. Mukonzo J, Aklillu E, Marconi V, Schinazi RF. Potential drug–drug interactions between antiretroviral therapy and treatment regimens for multi-drug resistant tuberculosis: Implications for HIV care of MDR-TB co-infected individuals. Int J Infect Dis 2019;83:98–101. https://doi.org/10.1016/j.ijid.2019.04.009.

2. Overall 3 AEs are highlighted throughout the narrative – hearing loss, depression and renal impairment. However, looking at table S9, there are several other adverse events for which RR was greater than those for these adverse events. There is also no mention of whether statistical significance of these associations was checked. Hence, the reason for prioritizing the mention of just 3 adverse events needs to be clarified.

Thank you very much for the feedback. Regarding the critique to provide reasons for prioritizing 3 AEs instead of all AEs, we believe that out of all analyses for specific AEs, only hearing loss, depression, and renal impairment yielded statistically significant results (i.e. confidence interval [CI] does not cross the reference line). We utilized CIs instead of p-values for point estimates as p-values may be misleading especially when only small studies or sample sizes are included in the analysis. This is especially true considering that p-values are usually interpreted according to a threshold (i.e. 0.05), which is arbitrary and may yield insufficient statistical power in some circumstances, including in the setting of small meta-analyses[1]. Hence, we believe that CIs may provide better statistical interpretation than p-values especially in the context of meta-analyses.

Reference:

Schünemann HJ, Vist GE, Higgins JPT, Santesso N, Deeks JJ, Glasziou P, Akl EA, Guyatt GH. Chapter 15: Interpreting results and drawing conclusions. In: Higgins JPT, Thomas J, Chandler J, Cumpston M, Li T, Page MJ, Welch VA, editors. Cochrane Handbook for Systematic Reviews of Interventions version 6.1 (updated September 2020). Cochrane, 2020. Available from: www.training.cochrane.org/handbook

3. It may also be worth mentioning here that in Table 1 in the main narrative, where meta-analysis and sub group analyses for the pooled adjusted effects on the association between HIV co-infection and adverse events occurrence, is presented the RR for any AE is more than 1 for HIV+ individuals while RR for serious AE is equivocal for HIV+ and HIV- individuals. However, the three commonly associated AEs, as above, are generally considered serious AE (specifically hearing loss and renal impairment).

Thank you very much for the feedback. We agree with the reviewer that it is worth discussing that although our findings indicate that HIV was independently associated with AE occurrence in DR-TB patients, we also discovered that HIV was not associated with an increased risk of serious AE. That being said, a majority of AEs encountered in HIV/DR-TB patients are manageable and not life-threatening, thus further indicating that antiretroviral treatment should not be deferred in such patients.

"Altogether, although we discovered that HIV co-infection was associated with an increased risk of developing any AE in DR-TB patients, ART should not be needlessly deferred in such patients. This is especially true considering that HIV co-infection was not associated with an increased risk of serious AE occurrence. Rather, we encourage clinicians to increase pharmacovigilance on HIV/DR-TB co-infected patients, especially in terms of ototoxicity, nephrotoxicity, and depressive symptoms. Therefore, routine audiological, laboratory (i.e. renal panel), and mental health assessments on such patients are strongly recommended." [see page 18 line 331-334]

On the other hand, although hearing loss and renal impairment may be debilitating and considered severe, we defined serious AE in this study according to the criteria published by the World Health Organization, including AEs leading to treatment discontinuation, requiring prolongation or initiation of hospitalization, requiring immediate interventions to prevent permanent damage, or resulting in significant disabilities, congenital abnormalities, or death[1]. We have also added the operational definition of serious AE used in this study in the methods section. 

"A serious AE was defined as an AE leading to treatment suspension, withdrawal or discontinuation, requiring prolonged hospitalization or immediate interventions to prevent permanent damage, or resulting in significant disabilities, congenital abnormalities, or death.[1]" [see page 6-7 line 124-127]

References:

World Health Organization. WHO consolidated guidelines on drug-resistant tuberculosis treatment 2019. Geneva: World Health Organization; 2019

4. It is also expected that ‘serious AE’ would be a subset of ‘any AE’. However, this doesn’t appear to be the case in analysis shown in Table 1. This needs to be clarified.

Thank you very much for the feedback. Regarding the query concerning the difference between ‘Serious AE’ subgroup and ‘Any AE’ subgroup, some studies (i.e. Merid et al[1], van der Walt et al[2], and Kvasnovsky et al[3]) only reported SAEs rather than any AEs. Hence, we decided to perform a subgroup analysis based on the reporting criteria (i.e. any AE vs serious AE only) to investigate whether the pooling of these studies to the main model introduced heterogeneity or significantly affected the estimate. From the analysis, we discovered that incorporating the studies reporting only SAEs did not significantly affect the point estimate, nor did it introduce substantial heterogeneity (see Table 1).

References:

1. Merid MW, Gezie LD, Kassa GM, Muluneh AG, Akalu TY, Yenit MK. Incidence and predictors of major adverse drug events among drug-resistant tuberculosis patients on second-line anti-tuberculosis treatment in Amhara regional state public hospitals; Ethiopia: a retrospective cohort study. BMC Infect Dis 2019;19:286. https://doi.org/10.1186/s12879-019-3919-1.

2. Van der Walt M, Lancaster J, Odendaal R, Davis JG, Shean K, Farley J. Serious Treatment Related Adverse Drug Reactions amongst Anti-Retroviral Naïve MDR-TB Patients. PLoS One 2013;8. https://doi.org/10.1371/journal.pone.0058817.

3. Kvasnovsky CL, Cegielski JP, Erasmus R, Siwisa NO, Thomas K, der Walt ML van. Extensively Drug-Resistant TB in Eastern Cape, South Africa: High Mortality in HIV-Negative and HIV-Positive Patients. JAIDS J Acquir Immune Defic Syndr 2011;57:146–52. https://doi.org/10.1097/QAI.0b013e31821190a3.

5. Terms ‘adverse reaction’ and ‘adverse events’ seem to have been used interchangeably at some places. The former term may be used if causality assessment has been conducted.

Thank you very much for the feedback. We have rechecked the utilization of the term ‘adverse reaction’ and ‘adverse events’ and changed them according to the context.

"This alarming evidence is further aggravated by the fact that DR-TB patients are more susceptible to drug-related adverse events (AEs) when compared to drug-susceptible TB patients." [see page 4 line 58-60]

"This meta-analysis revealed that HIV co-infection was independently associated with AEs in DR-TB patients" [see page 16 line 276-277]

"In conclusion, this meta-analysis adds to the growing body of evidence supporting the independent association between HIV co-infection and AEs in DR-TB patients." [see page 19 line 364-365]

6. Line 54 – ‘440,000 new cases and 150,000 deaths each year’. These figures are for RR/MDR-TB while the sentence starts with only DR-TB. DR-TB is an encompassing term for other forms/types of resistance as well.

Thank you very much for the feedback. Regarding the critique to clarify the epidemiological measures of DR-TB, we have revised the figures, breaking down DR-TB into rifampicin-resistant- and isoniazid-resistant-TB. In addition, we have also updated the data according to the latest report.

"Drug-resistant tuberculosis (DR-TB), defined as the emergence of resistance pattern of TB bacilli to one or more anti-TB drugs[1], remains a major global health burden with approximately 465,000 cases of rifampicin-resistant/multidrug-resistant tuberculosis (RR/MDR-TB) and 1.4 million cases of isoniazid-resistance TB cases in 2019[2]. Moreover, about 182,000 deaths from RR/MDR-TB were also reported in the same year.[3]" [see page 4 line 54-58]

We believe that these numbers represent the prevalence of DR-TB as a whole. This is saliently true considering that these figures include multidrug-resistant, extensively drug-resistant, polydrug-resistant, and total drug-resistant TB patients who had resistance to either rifampicin or isoniazid.

References:

1. World Health Organization. WHO consolidated guidelines on drug-resistant tuberculosis treatment 2019. Geneva: World Health Organization; 2019

2. Singh A, Prasad R, Balasubramanian V, Gupta N. Drug-resistant tuberculosis and HIV infection: current perspectives. HIV/AIDS (Auckland) 2020;12:9–31.

3. Institute of Medicine. Facing the reality of drug-resistant tuberculosis in India: Challenges and potential solutions - Summary of a joint workshop by the Institute of Medicine, the Indian National Science Academy, and the Indian Council of Medical Research. Washington (DC): National Academies Press; 2012.

7. Line 93-94 – ‘AEs were defined as any untoward event occurring following the administration of second-line anti-TB agents’. Please clarify is this also included AEs associated with ARV administration.

Thank you very much for the feedback. Regarding the query to clarify the definition of AE, we have clarified that the definition includes both AEs from second-line anti-TB agents and antiretroviral agents.

"AEs were defined as any untoward event occurring following the administration of second-line anti-TB and/or antiretroviral (ARV) agents." [see page 5 line 100-101]

8. Table 1: ‘HR’ and ‘OR’ abbrevi-ations in the ‘Outcome’ column need to be elaborated.

Thank you very much for the feedback. We have add-ed the definition of the abbreviation ‘HR’ and ‘OR’ in the footnote of Table 1.

AE, adverse event; CI, confidence interval; ES, ef-fect size; HIV, human immunodeficiency virus; HR, hazard ratio; MDR-TB, multidrug-resistant tubercu-losis; N, total sample size; NA, not available; OR, odds ratio; RR, risk ratio; TB, tuberculosis; XDR-TB, extensively drug-resistant tuberculosis [see the footnote of Table 1 in page 14 line 237-240]

9. As per Table 1, the RR for adverse events seems more pronounced for XDR-TB patients but this doesn’t find mention in subsequent narrative (line 234).

Thank you very much for the feedback. Regarding the query on the RR for MDR-TB and XDR-TB, we judged that there was no significant difference between MDR-TB and XDR-TB from the analysis of the adjusted outcomes. Although the RR for XDR-TB was larger (vs MDR-TB: 1.61 vs 1.28; see Table 1), the 95% confidence interval (CI) for XDR-TB was also wider than MDR-TB (0.92-2.84 vs 0.98-1.66; see Table 1), thus suggesting uncertainties in the point estimate. Furthermore, data on the adverse events for XDR-TB are lacking since only one study (i.e. Olayanju et al[1]) was included in the subgroup analysis. Hence, we decided to interpret the findings from unadjusted outcomes, which showed that the risks were more pronounced in MDR-TB than XDR-TB (RR [95% CI]: 1.22 [1.04-1.42] vs 0.80 [0.51-1.26]). Although the number of studies included in each subgroup was unbalanced (i.e. MDR-TB vs XDR-TB: 15 vs 4 studies), we believe that these 4 studies (i.e. Olayanju et al[1], Padayatchi et al[2], Shean et al[3], and Kvasnovsky et al[4]) may be more representative than the only study in the adjusted outcomes (i.e. Olayanju et al[1]). Nonetheless, we realized that unadjusted outcomes are limited by the fact that they are non-independent of potential confounders. Therefore, we decided to regard this as a limitation of our study.

"This study has several limitations. Although our findings highlighted the unfavorable effect of HIV co-infection on AE development in patients receiving DR-TB treatments, some of the pooled estimates were unadjusted for confounders due to study scarcity, emphasizing caution in the interpretation of our findings." [see page 18-19 line 348-351]

References:

1. Olayanju O, Esmail A, Limberis J, Gina P, Dheda K. Linezolid interruption in patients with fluoroquinolone-resistant tuberculosis receiving a bedaquiline-based treatment regimen. Int J Infect Dis 2019;85:74–9. https://doi.org/10.1016/j.ijid.2019.04.028.

2. Padayatchi N, Gopal M, Naidoo R, Werner L, Naidoo K, Master I, et al. Clofazimine in the treatment of extensively drug-resistant tuberculosis with HIV coinfection in South Africa: A retrospective cohort study. J Antimicrob Chemother 2014;69:3103–7. https://doi.org/10.1093/jac/dku235.

3. Shean K, Streicher E, Pieterson E, Symons G, van Zyl Smit R, Theron G, et al. Drug-Associated Adverse Events and Their Relationship with Outcomes in Patients Receiving Treatment for Extensively Drug-Resistant Tuberculosis in South Africa. PLoS One 2013;8. https://doi.org/10.1371/journal.pone.0063057.

4. Kvasnovsky CL, Cegielski JP, Erasmus R, Siwisa NO, Thomas K, der Walt ML van. Extensively Drug-Resistant TB in Eastern Cape, South Africa: High Mortality in HIV-Negative and HIV-Positive Patients. JAIDS J Acquir Immune Defic Syndr 2011;57:146–52. https://doi.org/10.1097/QAI.0b013e31821190a3.

10. In addition to monitoring for AEs, use of injection free regimen could be part of the recommendations because some of the common and serious AEs are associated with the injections being used for treatment of RR/MDR-TB

Thank you very much for the feedback. We agree with the reviewer that the use of injection-free DR-TB regimens should be preferred whenever possible. This is further supported by our findings where injectable-containing DR-TB regimens were associated with a more apparent risk than all-oral DR-TB regimens (RR 1.11 [95% CI: 1.01-1.21] vs RR 1.61 [95% CI: 0.92-2.84]; Table 1). Furthermore, injectable DR-TB agents (i.e. amikacin, kanamycin, and capreomycin) have also been proven to augment the risk of hearing loss and nephrotoxicity in DR-TB patients, which is also in line with our findings where aminoglycosides were among the most utilized drugs in the included studies. Lastly, Lan et al. further corroborated our premises by stating that these injectable agents were among the drugs with the highest risk of developing adverse events leading to permanent drug discontinuation.[1] Therefore, we have added a recommendation suggesting the widespread implementation of all-oral DR-TB regimens as the mainstay of DR-TB treatments. 

"This meta-analysis further indicated that the effect of HIV infection was more prominent in the increased risk of developing hearing loss and nephrotoxicity. It has been well-established that some anti-TB drugs, particularly aminoglycosides, are associated with hearing loss.[2,3] Aminoglycosides were also associated with nephrotoxicity as they were primarily renally excreted and may cause tubular necrosis.[4,5] This is in line with our findings where amikacin and kanamycin were among the most utilized drugs in the included studies, thus further elaborating the observed link. In addition, Lan et al. also stated that these drugs resulted in the highest incidence of adverse events leading to permanent drug discontinuation[1], indicating that a safer and better-tolerated regimen is required to reduce aminoglycoside-related morbidities. In this regard, all-oral treatment regimens become the preferred option for most DR-TB patients.[6] This is saliently true considering that our findings suggest that injectable-containing DR-TB regimens may result in a more apparent risk of developing AE, although further studies are required to confirm these premises. Furthermore, all-oral DR-TB regimens have shown to be more cost-effective with less logistical challenges[7,8], thus rendering them more worthwhile to be implemented especially in resource-limited settings. All in all, these necessitates the widespread implementation of oral-only regimens as the mainstay of DR-TB treatment." [see page 17 line 300-316]

References:

1. Lan Z, Ahmad N, Baghaei P, Barkane L, Benedetti A, Brode SK, et al. Drug-associated adverse events in the treatment of multidrug-resistant tuberculosis: an individual patient data meta-analysis. Lancet Respir Med 2020;8:383–94.

2. Hong H, Budhathoki C, Farley JE. Increased risk of aminoglycoside-induced hearing loss in MDRTB patients with HIV coinfection. Int J Tuberc Lung Dis 2018;22:667–74. https://doi.org/10.5588/ijtld.17.0830.

3. Seddon JA, Godfrey-Faussett P, Jacobs K, Ebrahim A, Hesseling AC, Schaaf HS. Hearing loss in patients on treatment for drug-resistant tuberculosis. Eur Respir J 2012;40:1277–86.

4. Updated guidelines on managing drug interactions in the treatment of HIV-related tuberculosis. MMWR Morb Mortal Wkly Rep. 2014;63:272.

5. Mingeot-Leclercq MP, Tulkens PM. Aminoglycosides: Nephrotoxicity. Antimicrob Agents Chemother 1999;43:1003–12

6. World Health Organization. WHO consolidated guidelines on drug-resistant tuberculosis treatment 2019. Geneva: World Health Organization; 2019

7. Ionescu AM, Mpobela Agnarson A, Kambili C, Metz L, Kfoury J, Wang S, et al. Bedaquiline- versus injectable-containing drug-resistant tuberculosis regimens: a cost-effectiveness analysis. Expert Rev Pharmacoeconomics Outcomes Res 2018;18:677–89

8. Reuter A, Tisile P, Von Delft D, Cox H, Cox V, Ditiu L, et al. The devil we know: is the use of injectable agents for the treatment of MDR-TB justified? Int J Tuberc Lung Dis. 2017;21:1114–26. 

Responses to Reviewer 2

1. On the conversion of OR/HR to RR. I am not convinced this method is appropriate for pooling effect estimates in a meta-analysis. This conversion method may still introduce bias. A sensitivity analysis where original odds ratios are pooled separately from risk ratios may be helpful to see if the pooled effects are similar across the effect measures used. In addition, it would be useful to indicate on any forest plots, in which they appear, the studies that have had their OR/HR converted.

Thank you very much for the feedback. We agree with the reviewer that the conversion of OR and HR to RR warrants cautions in interpreting our results, as these conversions may introduce heterogeneity to the model[1]. We decided to set our common measure as relative risk (RR) in order to ease the interpretation of our results, as the pooling of HR, OR, and RR separately may confuse and complicate the interpretation. Furthermore, to the best of our knowledge, there are no currently known method to convert HR to OR and vice versa. We have also elaborated in the Supplementary Material pg. 12 that the conversion of HR to RR may introduce bias ratio of at most 16% for outcome probability of 0.2-0.8, 45% for outcome probability of 0.1-0.9, and 93% for outcome probability of 0.05-0.95, which was remarkably lower than interchangeably using HR and RR which may introduce bias with a factor of 1.80 (80%) for outcome probability of 0.2-0.8, 2.47 (147%) for outcome probability of 0.1-0.9, and 19.00 (1900%) for outcome probability of 0.05-0.95. We have also added that only one study[2] reported outcome probability between 0.9-0.95, thus minimizing the risk of potential bias arising from these conversions.

Nonetheless, to anticipate bias and heterogeneity arising from these conversions, we decided to perform a subgroup analysis based on the original effect measure (i.e. OR and HR), which analysis showed that the heterogeneity arising from the pooling of these two converted effect measures was negligible. Although the point estimates for OR subgroup is substantially higher than those in HR subgroup (RR 1.15 [95% CI: 1.03-1.28] vs RR 1.05 [95% CI: 0.88-1.24]; see Table 1), we believe that this does not suggest potential bias arising from the conversion method as these numbers reflect specific study outcomes and are highly affected by other factors (i.e. study design, population, intervention, etc.)

Reference:

1. Riley RD, Moons KGM, Snell KYIE, Ensor J, Hooft L, Altman DG, et al. A guide to systematic review and meta-analysis of prognostic factor studies. BMJ. 2019;364:k4597

2. Smith JP, Gandhi NR, Shah NS, Mlisana K, Moodley P, Johnson BA, et al. The Impact of Concurrent Antiretroviral Therapy and MDR-TB Treatment on Adverse Events. JAIDS J Acquir Immune Defic Syndr 2020;83:47–55. https://doi.org/10.1097/QAI.0000000000002190.

2. One of the main predictors for AE are the specific drugs used (as mentioned in the discussion). Some stratification for cohorts that used PZA, Eto, and Km vs. those who did not (if possible given the data) should be attempted. As individual adjustments for drug use are not possible, this may help determine whether HIV is independently associated with AE rather than driven by the drugs used in specific cohorts.

Thank you very much for the feedback. We agree with the reviewer that it is imperative to investigate whether the effect of HIV on the development of AE in DR-TB patients was driven by HIV-related immunosuppression or drug-drug interactions. However, as we did not intend to analyze patient-level data, and as the treatment regimens used in each study intertwined with each other (e.g., studies not utilizing kanamycin utilized other aminoglycosides, i.e., amikacin or kanamycin), we were unable to analyze specific drug-drug interactions. Hence, we discussed for a way to explore the potential interplay of specific regimens and drug-drug interactions and decided to perform a subgroup analysis based on DR-TB treatment regimen, dichotomizing studies into studies utilizing injectable agents (i.e. kanamycin, amikacin, and capreomycin) and all-oral regimens (i.e. using bedaquiline, pretomanid, and linezolid). We discovered that injectable-containing DR-TB regimens was associated with a more apparent risk of developing adverse event (RR 1.11 [95% CI: 1.01-1.21] vs all-oral regimens: RR 1.61 [95% CI: 0.92-2.84]; Table 1), although it is worth noting that further research are required to confirm our findings as there is currently limited evidence on studies utilizing all-oral DR-TB regimens. Considering our findings, we have added a discussion favoring over the use of all-oral treatment regimens.

"In this regard, all-oral treatment regimens become the preferred option for most DR-TB patients.[1] This is saliently true considering that our findings suggest that injectable-containing DR-TB regimens may result in a more apparent risk of developing AE, although further studies are required to confirm these premises. Furthermore, all-oral DR-TB regimens have shown to be more cost-effective with less logistical challenges[2,3], thus rendering them more worthwhile to be implemented especially in resource-limited settings. All in all, these necessitates the widespread implementation of oral-only regimens as the mainstay of DR-TB treatment." [see page 16 line 290-299]

In addition, our findings also indicated that the effect of HIV was primarily driven by drug-drug interactions between ARV drugs and anti-DR-TB agents rather than HIV-related immunosuppression, as proven by subgroup analyses shown on Supplementary Figure S4 (ART vs not on ART: RR 1.21 [95% CI: 1.02-1.43] vs RR 1.24 [95% CI: 0.90-1.69]; CD4≥200 vs <200 cells/mm3: RR 0.94 [95% CI: 0.81-1.08] vs RR 0.96 [95% CI: 0.68-1.37]). In response to this, we have also added a discussion on the potential specific drug-drug interactions between ARV and anti-DR-TB drug.

"In addition, we also observed that HIV/DR-TB patients receiving concomitant ART were more vulnerable to AEs. Interestingly, we found that the extent of immunosuppression was not associated with AE occurrence, implying that the observed risk was presumably driven by drug-drug interactions rather than HIV-mediated immunosuppression. However, we were unable to explore the suspected drug-drug interactions as patient-level data were not available. Some anti-DR-TB drugs have been reported to induce adverse interactions when concomitantly administered with ART, including bedaquiline, delamanid, and moxifloxacin, which had documented interactions with several protease inhibitors (e.g. lopinavir/ritonavir) and non-nucleoside reverse transcriptase inhibitors (i.e. efavirenz and rilpivirine)[4], thus potentially leading to multiple toxicities resulting in deleterious health consequences[5]." [see page 17 line 309-316]

References:

1. World Health Organization. WHO consolidated guidelines on drug-resistant tuberculosis treatment 2019. Geneva: World Health Organization; 2019

2. Ionescu AM, Mpobela Agnarson A, Kambili C, Metz L, Kfoury J, Wang S, et al. Bedaquiline- versus injectable-containing drug-resistant tuberculosis regimens: a cost-effectiveness analysis. Expert Rev Pharmacoeconomics Outcomes Res 2018;18:677–89. https://doi.org/10.1080/14737167.2018.1507821.

3. Reuter A, Tisile P, Von Delft D, Cox H, Cox V, Ditiu L, et al. The devil we know: is the use of injectable agents for the treatment of MDR-TB justified? Int J Tuberc Lung Dis. 2017;21:1114–26

4. HIV Drug Interactions. Anti-tuberculosis treatment selectors. Univ Liverpool 2019. hiv-druginteractions.org/prescribing-resources (accessed October 11, 2020).

5. Mukonzo J, Aklillu E, Marconi V, Schinazi RF. Potential drug–drug interactions between antiretroviral therapy and treatment regimens for multi-drug resistant tuberculosis: Implications for HIV care of MDR-TB co-infected individuals. Int J Infect Dis 2019;83:98–101. https://doi.org/10.1016/j.ijid.2019.04.009.

6. World Health Organization. WHO consolidated guidelines on drug-resistant tuberculosis treatment 2019. Geneva: World Health Organization; 2019

7. Ionescu AM, Mpobela Agnarson A, Kambili C, Metz L, Kfoury J, Wang S, et al. Bedaquiline- versus injectable-containing drug-resistant tuberculosis regimens: a cost-effectiveness analysis. Expert Rev Pharmacoeconomics Outcomes Res 2018;18:677–89. https://doi.org/10.1080/14737167.2018.1507821.

8. Reuter A, Tisile P, Von Delft D, Cox H, Cox V, Ditiu L, et al. The devil we know: is the use of injectable agents for the treatment of MDR-TB justified? Int J Tuberc Lung Dis. 2017;21:1114–26

9. HIV Drug Interactions. Anti-tuberculosis treatment selectors. Univ Liverpool 2019. hiv-druginteractions.org/prescribing-resources (accessed October 11, 2020).

10. Mukonzo J, Aklillu E, Marconi V, Schinazi RF. Potential drug–drug interactions between antiretroviral therapy and treatment regimens for multi-drug resistant tuberculosis: Implications for HIV care of MDR-TB co-infected individuals. Int J Infect Dis 2019;83:98–101. https://doi.org/10.1016/j.ijid.2019.04.009.

3. In addition to the use of individual drugs, the use and availability of drugs has changed over the last two decades, as well as the guideline recommendations for DR-TB treatment. Combining cohorts from these different eras may not be appropriate as some assumptions between their comparability may be violated. Was the effect of HIV consistent between cohorts ending before 2010 and cohorts that began after 2010? Some stratification of cohort periods may provide some insight into this. If the data does not allow for these stratification to be made, some discussion regarding treatment periods in addition to the drugs used should be included.

Thank you very much for the feedback. We agree with the reviewer that the use of individual drugs may differ between studies. To the best of our knowledge, one substantial revolution in the treatment of DR-TB during the 2010s occurred in 2012 with the introduction of bedaquiline, delamanid, pretomanid, and linezolid, thus initiating the era of all-oral DR-TB treatment.[1] In light of this, we decided to add an additional analysis by dichotomizing studies based on their treatment regimens (i.e. injectable-containing regimens vs all-oral regimens). We discovered that injectable-containing DR-TB regimens were associated with a more apparent risk of developing AE (RR 1.11 [95% CI: 1.01-1.21] vs all-oral regimens: RR 1.61 [95% CI: 0.92-2.84]; Table 1), although further studies are required to confirm our findings as the current model was limited due to paucity of studies utilizing all-oral DR-TB regimens. In summary, only 4 studies utilized bedaquiline[2-5], and only two of which utilized and reported all-oral DR-TB regimens[2,4]. However, one[2] was excluded from the meta-analysis due to distinct study design (i.e. interventional study), thus leaving only one study[4] in the all-oral subgroup in the meta-analysis model. Considering our findings, we decided to add an elaboration on the use of all-oral DR-TB regimens in the discussion section.

"In this regard, all-oral treatment regimens become the preferred option for most DR-TB patients.[11] This is saliently true considering that our findings suggest that injectable-containing DR-TB regimens may result in a more apparent risk of developing AE, although further studies are required to confirm these premises. Furthermore, all-oral DR-TB regimens have shown to be more cost-effective with less logistical challenges[12,13], thus rendering them more worthwhile to be implemented especially in resource-limited settings. All in all, these necessitates the widespread implementation of oral-only regimens as the mainstay of DR-TB treatment." [see page 16 line 290-299]

As some studies do not allow the stratification of cohorts ending before 2010 and after 2010, we were unable to perform a stratification based on the treatment period. Nonetheless, we believe that the effect of HIV was consistent between studies as our model yielded negligible heterogeneity (I2=0%; p=0.75; Table 1). We hope that the additional subgroup analysis by DR-TB treatment regimens may fulfill the concerns raised by the reviewer.

References:

1. Swindells S. New drugs to treat tuberculosis. F1000 Med Rep. 2012;4: 12. doi:10.3410/M4-12

2. Conradie F, Diacon AH, Ngubane N, Howell P, Everitt D, Crook AM, et al. Bedaquiline, pretomanid and linezolid for treatment of extensively drug resistant, intolerant or non-responsive multidrug resistant pulmonary tuberculosis. N Engl J Med. 2020;382: 893–902. doi:10.1056/nejmoa1901814

3. Sineke T, Evans D, Schnippel K, van Aswegen H, Berhanu R, Musakwa N, et al. The impact of adverse events on health-related quality of life among patients receiving treatment for drug-resistant tuberculosis in Johannesburg, South Africa. Health Qual Life Outcomes. 2019;17: 94. doi:10.1186/s12955-019-1155-4

4. Olayanju O, Esmail A, Limberis J, Gina P, Dheda K. Linezolid interruption in patients with fluoroquinolone-resistant tuberculosis receiving a bedaquiline-based treatment regimen. Int J Infect Dis. 2019;85: 74–79. doi:10.1016/j.ijid.2019.04.028

5. Hughes J, Reuter A, Chabalala B, Isaakidis P, Cox H, Mohr E. Adverse events among people on delamanid for rifampicin-resistant tuberculosis in a high HIV prevalence setting. Int J Tuberc Lung Dis. 2019;23: 1017–1023. doi:10.5588/ijtld.18.0651

6. Hong H, Budhathoki C, Farley JE. Increased risk of aminoglycoside-induced hearing loss in MDRTB patients with HIV coinfection. Int J Tuberc Lung Dis. 2018;22: 667–674. doi:10.5588/ijtld.17.0830

7. Seddon JA, Godfrey-Faussett P, Jacobs K, Ebrahim A, Hesseling AC, Schaaf HS. Hearing loss in patients on treatment for drug-resistant tuberculosis. European Respiratory Journal. 2012. pp. 1277–1286. doi:10.1183/09031936.00044812

8. Updated Guidelines on Managing Drug Interactions in the Treatment of HIV-Related Tuberculosis. MMWR Morb Mortal Wkly Rep. 2014;63: 272. 

9. Mingeot-Leclercq MP, Tulkens PM. Aminoglycosides: Nephrotoxicity. Antimicrobial Agents and Chemotherapy. American Society for Microbiology; 1999. pp. 1003–1012. doi:10.1128/aac.43.5.1003

10. Lan Z, Ahmad N, Baghaei P, Barkane L, Benedetti A, Brode SK, et al. Drug-associated adverse events in the treatment of multidrug-resistant tuberculosis: an individual patient data meta-analysis. Lancet Respir Med. 2020;8: 383–394. doi:10.1016/S2213-2600(20)30047-3

11. World Health Organization. WHO consolidated guidelines on drug-resistant tuberculosis treatment. Geneva: World Health Organization; 2019. 

12. Ionescu AM, Mpobela Agnarson A, Kambili C, Metz L, Kfoury J, Wang S, et al. Bedaquiline- versus injectable-containing drug-resistant tuberculosis regimens: a cost-effectiveness analysis. Expert Rev Pharmacoeconomics Outcomes Res. 2018;18: 677–689. doi:10.1080/14737167.2018.1507821

13. Reuter A, Tisile P, Von Delft D, Cox H, Cox V, Ditiu L, et al. The devil we know: Is the use of injectable agents for the treatment of MDR-TB justified? Int J Tuberc Lung Dis. 2017;21: 1114–1126. doi:10.5588/ijtld.17.0468

4. Line 267 – this statement regarding what was reported in Lan et al. is not correct. In fact, what was reported for drug stoppage, if anything, was close to indicating the opposite (HIV may decrease the odds of having a drug stopped). This statement should be revised to reflect this.

Thank you very much for the feedback. We agree with the reviewer that we misinterpreted the statement by Lan et al. We realized that the statement referred to severe adverse events and not all adverse events, which may be incompatible to the context of our discussion. Furthermore, based on our findings in the Supplementary Figure S3, we did not discover any significant association between HIV seropositivity and severe AE (RR 1.23 [95% CI: 0.81-1.87]), which was in line with the findings by Lan et al. who stated that HIV status was not associated with AEs leading to drug discontinuation[1]. Hence, we decided to revise the statement as follows:

"This meta-analysis revealed that HIV co-infection was independently associated with adverse events in DR-TB patients. Although our findings were in contrast with the findings by Schnippel et al.[2], the study interpreted the association between HIV infection and drug-related AE using vote counting method based on subjective rules, thus potentially predisposing such analysis to poor performance validity[3]." [see page 16 line 276-290]

Reference:

1. Lan Z, Ahmad N, Baghaei P, Barkane L, Benedetti A, Brode SK, et al. Drug-associated adverse events in the treatment of multidrug-resistant tuberculosis: an individual patient data meta-analysis. Lancet Respir Med 2020;8:383–94. https://doi.org/10.1016/S2213-2600(20)30047-3.

2. Schnippel K, Firnhaber C, Berhanu R, Page-Shipp L, Sinanovic E. Adverse drug reactions during drug-resistant TB treatment in high HIV prevalence settings: A systematic review and meta-analysis. J Antimicrob Chemother 2017;72:1871–9. https://doi.org/10.1093/jac/dkx107.

3. Higgins J, Thomas J, Chandler J, Cumpston M, Li T, Page M, et al., editors. Cochrane handbook for systematic reviews of interventions version 6.0 (updated July 2019). 6th ed. Cochrane; 2019. https://doi.org/10.1002/9781119536604.

5. Issues of duration of TB drug use and time to adverse event are important for interpretation of effect estimates.

As mentioned above: Issues of duration of TB drug use and time to adverse event are important for interpretation of effect estimates, and there should be some mention of this in the discussion.

Thank you very much for the feedback. We agree with the reviewer that the duration of DR-TB treatment and the onset of adverse events (AEs) are imperative to take account upon interpreting our findings. This is particularly important in order to make a suggestion on the optimal timing of treatment monitoring, thus allowing the early detection and prompt treatment of such AEs to take place. However, we were unable to perform such an analysis due to differences in the length and time point of follow-up between the included studies. Hence, we decided to discuss this matter qualitatively as shown below.

"Altogether, although we discovered that HIV co-infection was associated with an increased risk of developing any AE in DR-TB patients, ART should not be needlessly deferred in such patients. This is especially true considering that HIV co-infection was not associated with an increased risk of serious AE occurrence. Rather, we encourage clinicians to increase pharmacovigilance on HIV/DR-TB co-infected patients, especially in terms of ototoxicity, nephrotoxicity, and depressive symptoms. Therefore, routine audiological, laboratory (i.e. renal panel), and mental health assessments on such patients are strongly recommended. These routine assessments should be performed periodically by taking into account the common onset of each AEs. However, our current data did not permit such an analysis due to differences in follow-up duration, thus limiting our ability to explore these factors. According to a study by Zhang et al.[1], most AEs in patients receiving injectable-containing DR-TB regimens occurred within the first six months. In contrast, most AEs in all-oral regimens appeared to develop more quickly, ranging between two weeks to three months[2,3]. Considering this, it is plausible for such assessments to be performed monthly, thus allowing the early detection and prompt management of potential AEs.[4] This is particularly important as AEs were among the most common reasons leading to treatment non-adherence and failure in DR-TB patients.[5]" [see page 337-347]

In addition, we have also added a recommendation to standardize the reporting standard of follow-up duration to

investigate the effects of duration of DR-TB treatments and onset of AEs.

Furthermore, we were unable to perform subgroup analysis by duration of follow-up due to heterogeneity in reporting, suggesting that a standardized reporting of follow-up duration in future studies are urgently needed. [see page 19 line 354-356]

References:

1. Zhang Y, Wu S, Xia Y, Wang N, Zhou L, Wang J, et al. Adverse events associated with treatment of multidrug-resistant tuberculosis in China: An ambispective cohort study. Med Sci Monit. 2017;23: 2348–2356. doi:10.12659/MSM.904682

2. Hughes J, Reuter A, Chabalala B, Isaakidis P, Cox H, Mohr E. Adverse events among people on delamanid for rifampicin-resistant tuberculosis in a high HIV prevalence setting. Int J Tuberc Lung Dis. 2019;23: 1017–1023. doi:10.5588/ijtld.18.0651

3. Olayanju O, Esmail A, Limberis J, Gina P, Dheda K. Linezolid interruption in patients with fluoroquinolone-resistant tuberculosis receiving a bedaquiline-based treatment regimen. Int J Infect Dis. 2019;85: 74–79. doi:10.1016/j.ijid.2019.04.028

4. Mase SR, Chorba T. Treatment of Drug-Resistant Tuberculosis. Clinics in Chest Medicine. W.B. Saunders; 2019. pp. 775–795. doi:10.1016/j.ccm.2019.08.002

5. Wang Y, Chen H, Huang Z, McNeil EB, Lu X, Chongsuvivatwong V. Drug non-adherence and reasons among multidrug-resistant tuberculosis patients in Guizhou, China: A cross-sectional study. Patient Prefer Adherence. 2019;13: 1641–1653. doi:10.2147/PPA.S219920

6. Line 57 – ‘Adverse event’ is first mentioned, but not abbreviated until line 64

Thank you very much for the feedback. We have revised the usage of the initial abbreviation of adverse event accordingly.

"This alarming evidence is further aggravated by the fact that DR-TB patients are more susceptible to drug-related adverse events (AEs) when compared to drug-susceptible TB patients[1], indicating that better understanding on the factors associated with the development of AEs during DR-TB treatment is urgently needed. This is saliently important, considering that AEs remain as one of the major predictor of unfavorable treatment outcomes.[2,3]" [see page 4 line 58-63]

References:

1. Wu S, Zhang Y, Sun F, Chen M, Zhou L, Wang N, et al. Adverse events associated with the treatment of multidrug-resistant tuberculosis: A systematic review and meta-analysis. Am J Ther 2016;23:e521--30. https://doi.org/10.1097/01.mjt.0000433951.09030.5a.

2. Shean K, Streicher E, Pieterson E, Symons G, van Zyl Smit R, Theron G, et al. Drug-Associated Adverse Events and Their Relationship with Outcomes in Patients Receiving Treatment for Extensively Drug-Resistant Tuberculosis in South Africa. PLoS One 2013;8. https://doi.org/10.1371/journal.pone.0063057.

3. Sineke T, Evans D, Schnippel K, van Aswegen H, Berhanu R, Musakwa N, et al. The impact of adverse events on health-related quality of life among patients receiving treatment for drug-resistant tuberculosis in Johannesburg, South Africa. Health Qual Life Outcomes 2019;17:94. https://doi.org/10.1186/s12955-019-1155-4.

7. Materials/methods – was a protocol registered/published?

Thank you very much for the feedback. We have added our protocol to the text, which was prospectively registered in PROSPERO on 5 July 2020.

"This review was conducted based on the guideline of systematic review of prognostic factor studies guideline proposed by Riley et al.[1] and was reported according to the Preferred Reporting Items for Systematic Reviews and Meta-Analyses (PRISMA) statement[2]. A detailed protocol has been prospectively registered in PROSPERO (CRD42020185029[3]). Deviations from the protocol are described in Supplementary Table S1." [see page 4-5 line 76-81]

In addition to citing our protocol, we have also specified the protocol in the PRISMA checklist (see Supplementary Material_PRISMA checklist). Furthermore, we have also added a supplementary table, listing all deviations made along with the explanations (see Supplementary Table S1). Following a thorough review, we discovered that interventional studies were not a criterion for exclusion, thus we decided to include the interventional study in our systematic review. The details of the revision are illustrated in Figure 1. As there is only one interventional study, we decided to exclude the study from the quantitative analysis (i.e., meta-analysis) and opted to analyze the study qualitatively. Details of the study’s characteristics and outcomes can be seen on Supplementary Table S6 and S9, respectively, while the result of risk of bias assessment of the aforementioned study are illustrated in Supplementary Figure S1. 

References:

1. Riley RD, Moons KGM, Snell KIE, Ensor J, Hooft L, Altman DG, et al. A guide to systematic review and meta-analysis of prognostic factor studies. BMJ. 2019;364. doi:10.1136/bmj.k4597

2. Moher D, Liberati A, Tetzlaff J, Altman DG, The PRISMA Group. Preferred reporting items for systematic reviews and meta-analyses: The PRISMA statement. PLoS Med. 2009;6: e1000097. doi:10.1371/journal.pmed.1000097

3. Lazarus G, Soetikno V, Iskandar A, Louisa M. The burden of human immunodeficiency virus infections on adverse events occurrence in the treatment of drug-resistant tuberculosis: a systematic review and meta-analysis. PROSPERO 2020. CRD42020185029. [cited 2021 Jan 19]. Available: https://www.crd.york.ac.uk/PROSPERO/display_record.php?RecordID=185029

8. Newcastle-Ottawa scale is not a good tool and for these study designs involving an intervention (or in general: see Stang et al. 2018 Eur J Epidemiol. “Case study in major quotation errors: a critical commentary on the Newcastle–Ottawa scale”). I think the quality assessment should be redone using ROBINS-I tool or similar to assess bias. Additionally, a sensitivity analysis stratifying on study quality as done with the NOS grading should be included as well – as selection bias, duration of treatment, and timing of AE may all affect the point estimates presented in the included studies. 

Thank you very much for the feedback. We agree with the reviewer that Newcastle-Ottawa Scale (NOS) may be limited due to poor inter-rater reliability.[1] Hence, prior to quality assessments, we performed an in-depth discussion among the investigators in order to align our comprehension and perceptions on the signaling questions in hope that our assessments will be sufficiently objective. Furthermore, most of the included studies were longitudinal observational studies with only one study each being an experimental[2] and a cross-sectional study[3], hence we believe that NOS is an appropriate tool to assess the quality of the respective studies. This is also in line with the recommendation by Zeng et al.[4] On the other hand, as for the experimental study, we decided to assess the quality of the study using the ROBINS-I tool as the study was single-armed and non-randomized. The results of the risk of bias assessment of the study can be seen in Supplementary Figure 1. The choice of the risk of bias tools was made prospectively as shown in our PROSPERO protocol (ID: CRD42020185029).[5]

References:

1. Oremus M, Oremus C, Hall GBC, McKinnon MC. Inter-rater and test–retest reliability of quality assessments by novice student raters using the Jadad and Newcastle–Ottawa Scales. BMJ Open. 2012;2:e001368

2. Conradie F, Diacon AH, Ngubane N, Howell P, Everitt D, Crook AM, et al. Bedaquiline, pretomanid and linezolid for treatment of extensively drug resistant, intolerant or non-responsive multidrug resistant pulmonary tuberculosis. N Engl J Med. 2020;382: 893–902. doi:10.1056/nejmoa1901814

3. Sineke T, Evans D, Schnippel K, van Aswegen H, Berhanu R, Musakwa N, et al. The impact of adverse events on health-related quality of life among patients receiving treatment for drug-resistant tuberculosis in Johannesburg, South Africa. Health Qual Life Outcomes. 2019;17: 94. doi:10.1186/s12955-019-1155-4

4. Zeng X, Zhang Y, Kwong JSW, Zhang C, Li S, Sun F, et al. The methodological quality assessment tools for preclinical and clinical studies, systematic review and meta-analysis, and clinical practice guideline: a systematic review. J Evid Based Med. 2015;8(1):2-10

5. Lazarus G, Soetikno V, Iskandar A, Louisa M. The burden of human immunodeficiency virus infections on adverse events occurrence in the treatment of drug-resistant tuberculosis: a systematic review and meta-analysis. PROSPERO 2020. CRD42020185029. [cited 2021 Jan 19]. Available: https://www.crd.york.ac.uk/PROSPERO/display_record.php?RecordID=185029

9. Further, was there an a priori adjustment set for which you would considered appropriate to pool adjusted effect estimates? A mix of estimated effects that may have been adjusted for different covariates may not be appropriate either (especially if they are being converted). Some clear mention of this should be included in the methods as well.

Thank you very much for the feedback. We agree with the reviewer that the incorporation of studies with diverse adjustment sets may be inappropriate as it may complicate the interpretation of our analysis. Hence, we have defined a minimum adjustment factors of age and sex as a criterion for studies to be included in the analysis of the adjusted outcome.

"Both adjusted and unadjusted estimates were pooled in the meta-analysis; however, adjusted estimates were prioritized for the interpretation of the results[1]. As the covariates adjusted in each study are highly variable[2], we pre-specified a minimum adjustment factors of age and sex for study estimates to be included in the analysis." [see page 7 line 144-147]

Furthermore, as this criterion was not specified in our protocol, we have also added an explanation to the deviation in Supplementary Table S1.

"We realized that it may be inappropriate to indifferently pool the adjusted outcomes of each study into a single model as the adjustment factors utilized may substantially differ, thus subsequently complicating the interpretation of our meta-analysis results[2]. Hence, we decided to set an additional inclusion criterion to the meta-analysis, in which a study must adjust for at least age and sex to be included in the quantitative analysis of adjusted outcomes." [see Supplementary Table S1]

References:

1. Dretzke J, Ensor J, Bayliss S, Hodgkinson J, Lordkipanidzé M, Riley RD, et al. Methodological issues and recommendations for systematic reviews of prognostic studies: an example from cardiovascular disease. Syst Rev 2014;3:140. https://doi.org/10.1186/2046-4053-3-140.

2. Riley RD, Moons KGM, Snell KIE, Ensor J, Hooft L, Altman DG, et al. A guide to systematic review and meta-analysis of prognostic factor studies. BMJ 2019;364. https://doi.org/10.1136/bmj.k4597.

10. Line 85 – “No language restrictions were applied”, but on line 96 an exclusion criteria is ‘articles not in English’, can you please clarify language restrictions?

Thank you very much for the feedback. Regarding the critique that there is a contradiction in the methods section concerning the eligibility criteria, we intended to assess whether language restrictions may affect our findings. Hence, we did not filter the search results by language, but rather excluded the potentially eligible non-English articles discovered during the search.

"Conversely, criteria for exclusion were: (1) non-original research, including qualitative research, case studies, reports, or case series with <20 patients; (2) irretrievable full-text articles; or (3) articles not in English." [see page 5-6 line 101-103]

The implementation of this criterion enabled us to assess the language bias arising from the language limitation, which we deemed negligible considering the large number of patients pooled in this systematic review and the fact that only three potentially eligible non-English articles were excluded. Regardless of this, we have acknowledged these as a limitation in the text as follows.

"Lastly, although language bias may arise from our eligibility criteria, our study included a relatively large number of patients and only three non-English articles were excluded, suggesting that any language bias may be negligible." [see page 19 line 356-359]

11. Line 120 – I am not sure what is meant, do you mean data was extracted/estimated from figures if unavailable from text/tables? Some clarification would be helpful.

Thank you very much for the feedback. Regarding the query to clarify the following statement “In the case of studies reporting the aforementioned outcomes only through graphical illustrations, the data were digitized with GetData Graph Digitizer ver. 2.26 (www.getdata-graph-digitizer.com).” [page 7 line 127-129], some studies (i.e. Figure 3 in Brust et al[1] and Figure 1-2 in Schnippel et al[2]) only reported the outcomes on adverse events through graphical illustrations (i.e. figures). Hence, in order to maximize the utilization of the available data, we decided to extract the data points of those figures by using a digitization software (i.e. GetData Graph Digitizer). To the best of our knowledge, such a practice is encouraged by Cochrane, especially in cases where the queried authors are unresponsive, and the data are not available elsewhere.[3]

References:

1. Brust JCM, Shah NS, Mlisana K, Moodley P, Allana S, Campbell A, et al. Improved Survival and Cure Rates with Concurrent Treatment for Multidrug-Resistant Tuberculosis-Human Immunodeficiency Virus Coinfection in South Africa. Clin Infect Dis 2018;66:1246–53. https://doi.org/10.1093/cid/cix1125.

2. Schnippel K, Berhanu RH, Black A, Firnhaber C, Maitisa N, Evans D, et al. Severe adverse events during second-line tuberculosis treatment in the context of high HIV Co-infection in South Africa: A retrospective cohort study. BMC Infect Dis 2016;16. https://doi.org/10.1186/s12879-016-1933-0.

3. Li T, Higgins JPT, Deeks JJ. Chapter 5: Collecting data. In: Higgins JPT, Thomas J, Chandler J, Cumpston M, Li T, Page MJ, Welch VA, editors. Cochrane Handbook for Systematic Reviews of Interventions version 6.1 (updated September 2020). Cochrane, 2020. Available from: www.training.cochrane.org/handbook

Additional changes

1. We changed the position of the ‘p-values’ in the ‘Outcome’ column in Table 1. The p-values reflect between-subgroup significance. Hence, we believe that it is more appropriate for these values to be placed in the same row with the ‘type of subgroup’ name rather than with one of the subgroups. 

2. We have also added funding information on the submission system. We discovered that we had not provided relevant financial disclosure during the initial submission process, while in fact our project received a grant from the Universitas Indonesia. Hence, we have added relevant information, stating that:

This project was funded by the PUTI Q1 research grant from the Universitas Indonesia (grant number: BA-1005/UN2.RST/PPM.00.03.01/2020). The funder had no role in study design, data collection and analysis, decision to publish, or preparation of the manuscript.

We have also inputted relevant information to the submission system, stating that the author VS was the recipient of the aforementioned grant.

We hope you will kindly reconsider our submission and we are looking forward to the publication of our manuscript. Thank you very much in advance for your kind consideration.

Sincerely yours,

Representing all authors

Gilbert Lazarus

Faculty of Medicine Universitas Indonesia, 

Jl. Salemba Raya No. 6, Jakarta 10430, Indonesia,

E-mail: gilbert.lazarus@ui.ac.id

---

## [Decision Letter · Decision Letter 1]

8 Feb 2021

PONE-D-20-36024R1

The effect of human immunodeficiency virus infection on adverse events during treatment of drug-resistant tuberculosis: a systematic review and meta-analysis

PLOS ONE

Dear Dr. Lazarus,

Thank you for submitting your manuscript to PLOS ONE. After careful consideration, we feel that it has merit but does not fully meet PLOS ONE’s publication criteria as it currently stands. Therefore, we invite you to submit a revised version of the manuscript that addresses the points raised during the review process.

We look forward to receiving your revised manuscript.

Kind regards,

Endi Lanza Galvão

Academic Editor

PLOS ONE

Reviewers' comments:

Reviewer's Responses to Questions

**Comments to the Author**

1. If the authors have adequately addressed your comments raised in a previous round of review and you feel that this manuscript is now acceptable for publication, you may indicate that here to bypass the “Comments to the Author” section, enter your conflict of interest statement in the “Confidential to Editor” section, and submit your "Accept" recommendation.

Reviewer #1: All comments have been addressed

Reviewer #2: All comments have been addressed

2. Is the manuscript technically sound, and do the data support the conclusions?

Reviewer #1: Yes

Reviewer #2: Yes

3. Has the statistical analysis been performed appropriately and rigorously? 

Reviewer #1: Yes

Reviewer #2: Yes

4. Have the authors made all data underlying the findings in their manuscript fully available?

Reviewer #1: Yes

Reviewer #2: No

5. Is the manuscript presented in an intelligible fashion and written in standard English?

Reviewer #1: Yes

Reviewer #2: Yes

6. Review Comments to the Author

Reviewer #1: It is stated in the response from authors that "the role of specific regimen and drug-drug interaction may provide essential information in determining which DR-TB treatment regimen to use. However, as we did not intend to analyze patient-level data, and as the treatment regimens used in each study intertwined with each other, we were unable to analyze specific drug-drug interaction". It would be good to include this in the section on limitations.

Reviewer #2: (No Response)

7. PLOS authors have the option to publish the peer review history of their article (what does this mean?). If published, this will include your full peer review and any attached files.

Reviewer #1: **Yes: **Vineet Bhatia

Reviewer #2: No

---

## [Author Response · Author response to Decision Letter 1]

16 Feb 2021

February 16th, 2021

Dear Prof. Joerg Heber, Dr. Endi Lanza Galvão, and respected reviewers,

Thank you for the comments from the editor and reviewers on our manuscript entitled “The ef-fect of human immunodeficiency virus infection on adverse events during treatment of drug-resistant tuberculosis: a systematic review and meta-analysis” by Gilbert Lazarus, Kevin Tjoa, Anthony William Brian Iskandar, Melva Louisa, Evans L. Sagwa, Nesri Padayatchi, Vivian Soetikno (manuscript ID: PONE-D-20-36024). We really appreciate the constructive and de-tailed feedback on our manuscript. We have revised the current submitted manuscript based on the reviewers’ feedback.

Reviewer 1

1. It is stated in the response from authors that "the role of specific regimen and drug-drug interaction may provide essential information in determining which DR-TB treatment regimen to use. However, as we did not intend to analyze patient-level data, and as the treatment regimens used in each study intertwined with each other, we were unable to analyze specific drug-drug interaction". It would be good to include this in the section on limitations.

Thank you very much for the feedback. We agree with the reviewer that the fact that we only pooled aggregate-level data may imply that some potentially important factors remained unexplored in our study. In this regard, we have added the aforementioned fact as our limitation.

This study has several limitations. Although our findings highlighted the unfavorable effect of HIV co-infection on AE development in patients receiving DR-TB treatments, some of the pooled estimates were unadjusted for confounders due to study scarcity, emphasizing caution in the interpretation of our findings. In addition, there is a need for further studies evaluating the independent effect of ART and CD4 count on AE occurrence in HIV/DR-TB patients, and studies comparing the effect of HIV infection across different spectrum of AE seriousness and severity. We were also unable to perform subgroup analysis by duration of follow-up due to heterogeneity in reporting, suggesting that a standardized reporting of follow-up duration in future studies are urgently needed. Furthermore, our study was also limited by the fact that some potentially important factors such as specific ARV/anti-DR-TB regimens and drug-drug interactions remained unexplored as patient-level data were not available. [page 17 line 357-359]

We hope you will kindly reconsider our submission and we are looking forward to the publication of our manuscript.

Thank you very much in advance for your kind consideration.

Sincerely yours,

Representing all authors

Gilbert Lazarus

Faculty of Medicine Universitas Indonesia, 

Jl. Salemba Raya No. 6, Jakarta 10430, Indonesia,

E-mail: gilbert.lazarus@ui.ac.id

---

## [Editor Report · Decision Letter 2]

18 Feb 2021

The effect of human immunodeficiency virus infection on adverse events during treatment of drug-resistant tuberculosis: a systematic review and meta-analysis

PONE-D-20-36024R2

Dear Dr. Lazarus,

We’re pleased to inform you that your manuscript has been judged scientifically suitable for publication and will be formally accepted for publication once it meets all outstanding technical requirements.

Kind regards,

Endi Lanza Galvão

Academic Editor

PLOS ONE
---

## [Editor Report · Acceptance letter]

22 Feb 2021

PONE-D-20-36024R2 

The effect of human immunodeficiency virus infection on adverse events during treatment of drug-resistant tuberculosis: a systematic review and meta-analysis 

Dear Dr. Lazarus:

I'm pleased to inform you that your manuscript has been deemed suitable for publication in PLOS ONE. Congratulations! Your manuscript is now with our production department. 

Kind regards, 

on behalf of

Dr. Endi Lanza Galvão 

Academic Editor

PLOS ONE